# Effects of Propolis on Infectious Diseases of Medical Relevance

**DOI:** 10.3390/biology10050428

**Published:** 2021-05-12

**Authors:** Nelly Rivera-Yañez, C. Rebeca Rivera-Yañez, Glustein Pozo-Molina, Claudia F. Méndez-Catalá, Julia Reyes-Reali, María I. Mendoza-Ramos, Adolfo R. Méndez-Cruz, Oscar Nieto-Yañez

**Affiliations:** 1Carrera de Médico Cirujano, Facultad de Estudios Superiores Iztacala, Universidad Nacional Autónoma de México, Tlalnepantla 54090, Estado de México, Mexico; nelly.rivera.yanez@iztacala.unam.mx (N.R.-Y.); glustein@iztacala.unam.mx (G.P.-M.); reali@unam.mx (J.R.-R.); merisam06@iztacala.unam.mx (M.I.M.-R.); armendez@unam.mx (A.R.M.-C.); 2División de Investigación y Posgrado, Facultad de Estudios Superiores Iztacala, Universidad Nacional Autónoma de México, Tlalnepantla 54090, Estado de México, Mexico; mendezcatalacf@ired.unam.mx; 3Facultad de Estudios Superiores Iztacala, Universidad Nacional Autónoma de México, Tlalnepantla 54090, Estado de México, Mexico; claudia_riveray@my.uvm.edu.mx; 4Laboratorio de Genética y Oncología Molecular, Laboratorio 5, Edificio A4, Facultad de Estudios Superiores Iztacala, Universidad Nacional Autónoma de México, Tlalnepantla 54090, Estado de México, Mexico; 5Laboratorio de Inmunología, Unidad de Morfofisiología y Función, Facultad de Estudios Superiores Iztacala, Universidad Nacional Autónoma de México, Tlalnepantla 54090, Estado de México, Mexico

**Keywords:** propolis, antibacterial, antifungal, antiparasitic, antiviral, bioactive compounds

## Abstract

**Simple Summary:**

Propolis is a beekeeping product with a complex and highly variable chemical composition. Many beneficial health properties have been reported. In this review, we will be focusing on compiling the studies carried out with propolis on infectious diseases of greater medical relevance. Likewise, the promises and challenges that propolis has to consolidate itself as a complementary therapy for the treatment of these diseases are analyzed.

**Abstract:**

Infectious diseases are a significant problem affecting the public health and economic stability of societies all over the world. Treatment is available for most of these diseases; however, many pathogens have developed resistance to drugs, necessitating the development of new therapies with chemical agents, which can have serious side effects and high toxicity. In addition, the severity and aggressiveness of emerging and re-emerging diseases, such as pandemics caused by viral agents, have led to the priority of investigating new therapies to complement the treatment of different infectious diseases. Alternative and complementary medicine is widely used throughout the world due to its low cost and easy access and has been shown to provide a wide repertoire of options for the treatment of various conditions. In this work, we address the relevance of the effects of propolis on the causal pathogens of the main infectious diseases with medical relevance; the existing compiled information shows that propolis has effects on Gram-positive and Gram-negative bacteria, fungi, protozoan parasites and helminths, and viruses; however, challenges remain, such as the assessment of their effects in clinical studies for adequate and safe use.

## 1. Introduction

Currently, most health systems around the world are based mainly on the prevention of diseases. The world is constantly exposed to a large number of pathogens that cause emerging and re-emerging disease. These pathogens differ widely in terms of severity and probability and have varying consequences for morbidity and mortality, jeopardizing not only health but also social and economic well-being. It is absolutely necessary to have a global health system that is able to prevent and respond effectively to the expanding and evolving infectious diseases, as well as solving an increasingly widespread antimicrobial resistance [1]. The need to prevent, identify, and respond to any infectious disease that compromises global health stability remains a national, regional, and international priority [2].

Existing natural products could be potential resources to find different compounds for the development of new drugs and relevant medicine [3], creating an area of study of great importance, since the immense difference of natural molecules could contribute bioactive compounds that help in therapeutic improvement [4]. Propolis is a natural resinous product elaborated by bees from material obtained from various botanical sources; it is mixed with bees’ wax and enzymes secreted by the bee’s salivary glands [5]. Characteristically, its composition is 50% resin, 30% wax, 10% essential oils, 5% pollen, and 5% other substances [6]. The propolis was informed to present about 300 distinct compounds [7]. The characteristic chemical groups identified in propolis are phenolic acids or their esters, flavonoids, terpenes, aromatic aldehydes and alcohols, fatty acids, stilbenes, and β-steroids [7,8]. In addition, both the biomedical effect and composition of propolis have a very high variability according to the region of collection, the surrounding plant sources, and the seasons [9,10]. Many reports have shown that propolis possesses antibacterial, antifungal, antiparasitic, antiviral, antioxidant, anti-inflammatory, antitumor, antidiabetic, and immunomodulatory properties [11,12,13,14,15,16,17,18,19]. Propolis is a bee product that contains a great variety of biomedical properties and a great spectrum of components that could be promising candidates for drug discovery, which could be used to treat characteristic affections of distinct diseases. Notably, infectious diseases are a public health problem, since they do not have adequate treatment because many pathogens have developed resistance to the different drugs used against them. This is where propolis and many other alternative and complementary medicine products play an important role, since they are easily accessible, allowing a high percentage of the world population to use them, providing options to complement current treatments. As such, it is necessary to clinically analyze the effectiveness of propolis to evaluate its potential in human health promotion.

## 2. Antibacterial Activity of Propolis

One of the main complications with diseases caused by bacteria is their resistance to the antibiotics commonly used against them. Antibiotics are chemical compounds that can act in two ways: inhibiting (bacteriostatic drugs) or killing (bactericidal drugs) bacteria. These drugs are characterized by a specific interaction with a defined target in the bacterial cell, and they are arguably the most important medical intervention introduced by humans [20]. Currently, the figures related to this problem are alarming: according to conservative numbers mentioned by the Centers for Disease Control (CDC), approximately 23,000 people are estimated to die annually only in the USA as a result of an infection with an antibiotic-resistant organism [21]. According to a report, antibiotic resistance is predicted to cause around 300 million premature deaths by 2050, with a loss of up to USD 100 trillion to the global economy [22]. Next, we address the main research describing the use and activities of propolis from different countries on some bacterial agents of greater medical relevance today.

### 2.1. Staphylococcus Infections

The genus *Staphylococcus* causes different infections in the human population like impetigo, scalded skin syndrome, toxic shock syndrome, pneumonia, endocarditis, and urinary tract infections, among others [23]. Some species of this genus are resistant to antibiotics, such as methicillin-resistant *Staphylococcus aureus* (MRSA) [24]. In this genus, we can highlight to *S. aureus* and *Staphylococcus epidermidis*.

Records exist that demonstrate the use of propolis since ancient civilizations, as it possesses a large number of biological properties, one of which is its antibacterial effect [9,25,26,27]. Currently, various investigations around the world have demonstrated the antibacterial capacity of different types of propolis; hence, various studies report that all the distinct varieties of propolis have different antibacterial activities [28]. On the American continent, propolis varies widely, each having different characteristics. In this context, the antibacterial activity of Canadian propolis was evaluated, which showed activity on *S. aureus* [29]. Likewise, the antibacterial effect of Brazilian propolis (red, green, and brown) from distinct areas was studied, with the authors finding that the red extracts demonstrated activity against different bacterial species, including *S. aureus*; however, the green and brown extracts showed less activity than red extracts [30]. Similarly, in Europe, French propolis demonstrated significant antibacterial activity against both methicillin-susceptible *Staphylococcus aureus* (MSSA) and MRSA [31]. Likewise, Polish propolis showed variability in its activity on twelve MSSA and MRSA clinical isolates [32].

In 2017, Al-Ani et al. mentioned the antibacterial activity of propolis of various geographic origins such as Germany, Ireland, and the Czech Republic. The three propolis samples showed moderate antibacterial effect on *S. aureus*, MRSA, and *S. epidermidis* [28]. Similarly, Italian propolis showed antibacterial activity on clinically isolated *S. aureus* and *S. epidermidis*. The propolis demonstrated an inhibition on lipase activity of 18 *Staphylococcus* spp. and an inhibition on the coagulase of 11 tested *S. aureus*. Propolis showed an inhibitory activity of the adhesion and consequent biofilm growth of *S. aureus* [33]. In another study, 53 propolis were obtained from various areas in Serbia, which revealed one type of blue propolis and one orange, depending on floral and geographical origin. Propolis samples showed an effect against different bacteria, including *S. aureus*, with the orange-type propolis samples showing higher antibacterial activity compared with the blue-type propolis samples (Table 1) [34]. The variety of climates and flora in Africa results in propolis with very particular characteristics; however, as with samples from the Americas and Europe, they showed an effect on strains of *S. aureus* and *S. epidermidis* [35]. Another study reported that Kenyan propolis showed differences in the antibacterial activity against *S. aureus* in three studied geographical areas [36]. In this context, we agree with the different authors who found a great variety of propolis that present a diversity of activity against Staphylococci in distinct regions around the world; these investigations have been important for the study of infections caused by this bacterial genus. However, in these studies, the chemical components of the propolis were not described, so the adequate and standardized use of propolis cannot yet be achieved [6].

As propolis functions to support the sterility and health of the beehive, the protective properties of the bioactive compounds in propolis can provide significant benefits for human health [37,38]. In this context, the flavonoids and esters of phenolic acids present in propolis are habitually the active components related to antibacterial effect [39]. Samples of distinct types of propolis from diverse regions in Brazil were studied (red, green, and brown), showing distinct antibacterial activities against different microorganisms, including *S. aureus*. Ferulic acid, gallic acid, caffeic acid, coumaric acid, *p*-coumaric acid, catechin, drupanin, kaempferide, artepillin C, luteolin, and pinocembrin have been identified in propolis, the researchers concluding that Brazilian propolis have various compounds, which present antibacterial activities that could be used for the elaborate of new medicines [40,41,42]. Similarly, the antibacterial activity of 20 Polish propolis obtained from distinct areas and 5 propolis from agricultural localities were studied. These samples showed distinct antibacterial activity toward *S. aureus* and *S. epidermidis*. For the 20 clinical isolates of *S. aureus* (16 MSSA and four MRSA), the propolis samples presented different activities, two of which showed higher antistaphylococcal activity, probably because these two samples contain more flavonoids that the other samples of propolis studied. The propolis originating from agricultural areas in Southern Poland presented a higher content of bioactive components (different flavonoids and phenolic acids). The samples of Polish propolis effectively eradicated staphylococcal biofilm, suggesting that the identified components are essential for the antibacterial effect of propolis [27,43]. Pinocembrin, galangin, and chrysin identified in South African propolis are known to possess antibacterial activity; combinations of these three flavonoids presented higher inhibition than flavonoids alone against different bacterial strains, including *S. aureus*. These flavonoids showed a synergistic effect to obtain a better antibacterial activity (Table 2) [44]. Propolis from Northern Morocco showed inhibitory effects against *S. aureus*, with the authors identifying different phenolic compounds such as caffeic acid, *p*-coumaric acid, ferulic acid, naringenin, pinocembrin, chrysin, galangin, pinobanksin, and quercetin [45]. We also agree with the studies that have reported the different propolis around the world presenting antistaphylococcal activity; these investigations have described the main active components of propolis and their activity against staphylococci, alone or in combination, which is of relevance for future research [44], since the search for more compounds and better combinations is necessary to treat infections occasioned by the *Staphylococcus* genus.

In another study, synergistic interactions were reported regarding combinations of Irish propolis and antibiotics (two-drug combinations: vancomycin, oxacillin, and levofloxacin) against different microbial pathogens, including MRSA. The authors concluded that the propolis from Ireland increased the synergistic effect and the effectiveness of antibiotics, mainly of vancomycin and oxacillin, that interact on cell-wall synthesis on drug-resistant bacteria [28]. In 2019, Grecka et al. observed the synergistic antistaphylococcal effect against *S. aureus* of one sample of Polish propolis combined with different drugs and fusidic acid; notably, all these drugs present an inhibitory action on protein synthesis [27]. Similarly, the activity of the combination of propolis from Poland with 10 antibiotics against staphylococci on *S. aureus* clinical isolates was proven, suggesting that the combinations of Polish propolis with different drugs potentiated the antibacterial effect on the various strains; however, no synergism was observed in the case of ciprofloxacin and chloramphenicol [32]. Likewise, the synergetic effect of propolis from Italy with some antibiotics on different bacterial strains was assessed, including *S. aureus* and *S. epidermidis*, reporting that Italian propolis enhanced the antibacterial activity of six different antibiotics [33].

Recently, Malaysian propolis and propolis nanoparticles (prepared with Malaysian propolis) exhibited antibacterial and antibiofilm properties against *S. epidermidis.* Propolis nanoparticles drastically inhibited biofilm growth by *S. epidermidis* and reduced the viability of biofilm bacteria compared with propolis extract. Propolis nanoparticles treatment showed significant disruption of biofilm and partial disruption by Malaysian propolis extract, decreasing bacteria in the biofilm. The gene expression in the tested bacteria described that genes related in intercellular adhesion (*IcaABCD, embp*) were downregulated by propolis nanoparticles. Propolis nanoparticles presented a synergistic effect with different drugs, suggesting efficient treatment. The authors concluded that propolis nanoparticles are more efficient than propolis extract alone in inhibiting bacterial biofilms by produce membrane alteration and reducing biofilm growth (Table 3) [46]. Notably, drug-resistant bacteria significantly affect health systems at present; in this context, studies with propolis and its potential antibacterial activity are promising. We agree with the different studies that have demonstrated the effectiveness of propolis, its bioactive components, and the combinations with various antibiotics against the *Staphylococcus* genus. These new sources of natural products are not aimed to replace antibiotic treatment but could be a complement in the treatment of these pathogens that now have resistance to antibiotics [47].

### 2.2. Streptococcus Infections

The genus *Streptococcus* is classified as Gram-positive and catalase-negative, appearing as cocci in pairs and chains on Gram stains. When grown on blood agar, they appear as small colorless colonies that cause beta or complete hemolysis [48]. The species of this genus are the cause of a large number of diseases in the human population, from acute to chronic infections with a wide array of manifestations in both adults and children [49]. In this genus, we can highlight *S. pyogenes, S. pneumoniae*, and *S. mutans*.

Three propolis samples of different geographic origins (Germany, Ireland, and Czech Republic) present moderate antibacterial effect on *S. pyogenes* and *S. pneumoniae* [28]. Similarly, Italian propolis showed antibacterial activity on different clinically isolated Gram-positive strains, including *S. pneumoniae* [33]. Similarly, a sample of Mexican propolis presented antibacterial activity against different microorganisms, including *S. mutans*; compounds such as pinocembrin, chrysin, galangin, alpinetin, dillenetin, isorhamnetin, ferulic acid, syringic acid, and caffeic acid were identified in the propolis. Several compounds (galangin, ferulic acid, syringic acid, and caffeic acid) also showed antibacterial activity against this oral pathogen [50]. In another study, the antibacterial effect of various samples of South Brazilian propolis was assessed: all showed activity against different bacterial strains, including *S. mutans*. All samples of propolis have an inhibitory action on *S. mutans* biofilm growth. In all these samples, diverse compounds were described, concluding that South Brazilian propolis could be an important resource of active components with properties for use in the pharmaceutical sector [42]. In similar research, the antibacterial and antibiofilm activities of propolis from Iran and its main compound, quercetin, were described on different bacterial strains, including *S. mutans* and *S. pneumoniae*, suggesting that Iranian propolis and quercetin were effective on the different bacteria studied and showed an inhibitory activity *S. mutans* biofilm adherence (Table 4) [51]. Several investigations have studied the activity of propolis and some of its bioactive compounds against the genus *Streptococcus*; although the results are encouraging, a limitation of these studies is that the possible mechanisms of action must be studied in vitro and in models [52], which would help to better understand this type of infection and its possible complementary treatments.

Interactions were reported regarding combinations of Irish propolis and distinct drugs (two-drug combinations: vancomycin, oxacillin, and levofloxacin) against different bacterial strains, including *S. pneumoniae* and *S. pyogenes*. The propolis from Ireland increased synergistic effect and the effectiveness of drugs that interact on cell-wall synthesis (vancomycin and oxacillin) [28].

### 2.3. Gastrointestinal Infections

Gastrointestinal infections constitute a great proportion of the acute and chronic disease burden in all the world. Some bacterial, viral, and parasitic microorganisms infect through contaminated food and water or from human to human. The WHO mentions that diarrhea causes 2.2 million deaths each year worldwide (about 4%) [53]. For this reason, the studies examining the effects of propolis on the main bacterial pathogens that cause gastrointestinal diseases are analyzed below.

The propolis of different geographic origins presented a moderate antibacterial effect on *Escherichia coli*, *Salmonella choleraesuis*, and *Shigella flexneri* [28]. In another study, 53 propolis were obtained from various areas of Serbia; the orange-type propolis samples showed higher antibacterial activity against *E. coli, Salmonella enteritidis, S. flexneri,* and *Listeria monocytogenes* [34]. Another study reported that Kenyan propolis showed differences in the antibacterial activity from three different geographical areas against different bacterial strains, including *E. coli* [36].

Similarly, Brazilian propolis (red, green, and brown; collected in diverse regions) as well as Southern Poland propolis showed distinct antibacterial activities against different microorganisms, including *E. coli* and *L. monocytogenes*; bioactive components such as ferulic acid, *p*-coumaric acid, caffeic acid, catechin, luteolin, drupanin, kaempferide, artepillin C, pinocembrin, chrysin, pinobanksin, apigenin, and kaempferol were identified [40,41,43]. In another study, pinocembrin, galangin, and chrysin (principal components South African propolis) were found to possess antibacterial activity against different microorganisms, including *L. monocytogenes* and *E. coli*, and combinations of these three flavonoids presented higher inhibition activity than components alone. The authors observed that these compounds worked synergistically to achieve the best antibacterial effect (Table 5) [44]. Propolis from Northern Morocco showed inhibitory effects against different Gram-negative strains, including *E. coli*; the researchers identified caffeic acid, *p*-coumaric acid, ferulic acid, naringenin, pinocembrin, chrysin, galangin, pinobanksin, and quercetin [45]. As mentioned by different authors who have studied the activity of propolis and some of its bioactive compounds on the effect against pathogens that cause gastrointestinal infections, we agree that the propolis present a great antibacterial diversity; nevertheless, these investigations contribute limited conclusions; therefore, it is necessary to carry out more studies focusing on understanding the antibacterial activity of propolis and trying to find its possible mechanism of action [54]. In addition, it is important to conduct in vivo and clinical trials in propolis of various areas to consider the differences in the chemical components of each one and, therefore, the different antibacterial activities that it may present [55].

Another research showed that Brazilian propolis presents a bacteriostatic effect on *Salmonella typhi*, and Bulgarian propolis presented a bactericidal effect and a synergism with chloramphenicol, tetracycline, and neomycin (act on the ribosome) on this same pathogen [56].

### 2.4. Nosocomial Infections

Nosocomial infections are not commonly found when admitted to hospital or are probably incubating. These infections are typically contracted in the hospitalization and generally manifest 48 h after [57]. The numbers of these infections are worrying: according to estimated figures from the CDC, in 2014, 11,282 patients suffered from healthcare-associated infections in USA hospitals alone. The main infections encompass primary bloodstream infection, surgical site infections, pneumonia, and urinary tract infections [58]. In this area, *Haemophilus influenza, Pseudomonas aeruginosa*, and *Klebsiella pneumoniae* are the cause of most of respiratory and renal nosocomial infections, respectively.

The antibacterial effect of Brazilian propolis (red, green, and brown) of various areas was studied; the red extracts demonstrated higher activity than green and brown extracts against different bacterial species, including *Klebsiella* sp. [30]. Similarly, a study reported the moderate antibacterial effect of propolis of distinct geographic regions (Germany, Ireland, and Czech Republic) on *P. aeruginosa, H. influenzae, K. pneumoniae*, and two clinical isolates of *K. pneumoniae* [28]. Another studies reported that Cameroonian, Congolese, and Kenyan propolis showed differences in antibacterial activity against various microorganisms, including *K. pneumoniae* and *P. aeruginosa* (Table 6) [35,36]. The greatest limitation of these investigations is that they remained at a qualitative level, only describing whether or not propolis presented activity; they did not mention any possible mechanism of action of propolis against these pathogens, which is essential to better understanding nosocomial infections and how to combat them [59,60].

Another study mentioned that various South Brazilian propolis showed activity against different bacterial strains, including *P. aeruginosa*. In all samples, gallic acid, caffeic acid, coumaric acid, artepillin C, and pinocembrin were identified [42]. Propolis originating from Southern Poland showed stronger antibacterial activity against different microorganisms, including *K. pneumoniae* and *P. aeruginosa*. Additionally, pinocembrin, chrysin, pinobanksin, apigenin, kaempferol, *p*-coumaric acid, ferulic acid, and caffeic acid were identified [43]. In other research, the effects of Albanian propolis were evaluated in various virulence factors of *P. aeruginosa*. Propolis inhibited the microbial development and biofilm growth; also, propolis decreased extracellular DNA release and phenazine production. Compounds were identified in the propolis, such as caffeic acid, *p*-coumaric acid, ferulic acid, isoferulic acid, quercetin, apigenin, pinobanksin, chrysin, pinocembrin, galangin, and caffeic acid phenethyl ester (CAPE), with the authors including that Albanian propolis contains different components with activity on biofilm-related infections [61]. In another investigation, combinations of pinocembrin, galangin, and chrysin (principal components of South African propolis) showed a better inhibitory effect than single compounds against different bacterial strains, including *P. aeruginosa* and *K. pneumoniae*, suggesting that these compounds present a synergistic interaction favoring antibacterial activity (Table 7) [44]. Likewise, propolis from Northern Morocco showed inhibitory effects against different Gram-negative strains, including *P. aeruginosa*. Different phenolic compounds, such as caffeic acid, *p*-coumaric acid, ferulic acid, naringenin, pinocembrin, chrysin, galangin, pinobanksin, and quercetin, were identified from the propolis [45]. We consider the studies on propolis and some of its bioactive compounds against pathogens that cause nosocomial infections to be of relevance, since they mentioned a possible mechanism of action [62], although more studies are needed related to this area. It is also necessary to carry out research using in vivo models and then to clinical trials to help knowledge possible via action of propolis and be able to combat this type of infection [63,64].

Another study reported that two-antibiotic combinations (vancomycin, oxacillin, and levofloxacin) and Irish propolis showed synergism on *H. influenzae*, concluding that the propolis from Ireland increases the synergism and effectiveness of vancomycin and oxacillin, which act on cell wall synthesis [28].

As we already mentioned, resistance to antibiotics is a serious health problem, since it makes it difficult to properly treat several diseases of bacterial origin. The documented effects of propolis and its derivatives on bacteria such as MRSA make them ideal candidates for clinical studies in order to evaluate their effectiveness on antbiotic-resistant bacterial diseases. The clinical application of propolis should not focus on the substitution of antibiotics, but on complementing and improving the efficacy of these when co-administered.

## 3. Antifungal Activity of Propolis

Fungal infections are responsible for over one million human deaths annually and are an increasingly important cause of mortality and morbidity [65]. In recent years, fungal infections have increased significantly, being considerably high in immunosuppressed patients [66]. Unfortunately, the low number of available treatments and the misuse of the antifungal medications have led to the selection of resistant microorganisms [67], which is why the search for new, effective, and inexpensive antifungal agents is crucial to overcoming existing resistance mechanisms [66]. Different natural products from distinct places and latitudes come to constitute a little-explored group of agents with antifungal capacity; of all these, propolis has special relevance [66], as recent studies have evaluated it as a natural product with potential for the development of antifungal drugs without toxicity [68,69].

### 3.1. Candidiasis

The genus *Candida* is a group of fungi known for their dimorphic capacity and is commonly isolated from the microbiome of healthy individuals (intestinal tract, oral cavity, skin, and vaginal cavity) [70,71,72]. However, when the host’s immunity becomes compromised by diseases such as HIV, AIDS, cytotoxic therapies, uncontrolled diabetes mellitus, or people of very young or very old age, *Candida* can behave like a pathogenic fungus. The progressive increase in the number of infections caused by *Candida* worldwide has increased in recent decades; this may be due to the significant increase in the population at risk, particularly the spread of HIV, immunosuppressive therapies, and the increase in the use of permanent devices [72,73,74]. The different species of *Candida* were classified as the fourth main agent that generate highly relevant infections worldwide; the magnitude of these diseases worldwide is alarming [72,75], and it has been recorded that infections caused by Candida species have a high crude mortality rate, exceeding the number caused by *S. aureus* and *P. aeruginosa* in nosocomial infections of the bloodstream [76]. It is important to highlight that the incidence in the annual rates of nosocomial infections of the bloodstream caused by Candida at the beginning of the century presented a variability of 6.0 to 13.3 and from 1.9 to 4.8 cases per 100,000 inhabitants in the United States and Europe, respectively [77,78,79]. Hence, *Candida* is a healthcare priority, and new antifungal therapeutic approaches are urgently needed. The propolis from different geographical regions has demonstrated anti-*Candida* activity, as described below.

The distinct clinical isolates of different species of the genus *Candida* extracted from vaginal exudates of patients with vulvovaginal candidiasis were completely suppressed by Brazilian propolis with a very small variation independent of the yeast species [80]. Likewise, Brazilian green propolis showed the ability to suppress the growth and biofilm formation of vaginal isolates of *C. albicans* [81]. In another research, the fungicidal effect of Brazilian propolis was demonstrated on three morphogenetic types of *C. albicans*, and the induced cell death was mediated by metacaspase and Ras signaling. This was corroborated by propolis inhibiting yeast transformation to hyphal growth. Moreover, a topically applied pharmaceutical formula based on propolis can partially control *C. albicans* infections in a vulvovaginal candidiasis infection in a mouse model [82].

Within Europe, the antifungal effect of different propolis has been investigated. The effect of four different Polish propolis samples on azole-resistant *Candida* clinical isolates was studied, with only one of the four propolis samples revealing high antifungal activity [83]. Similarly, Portuguese and French propolis presented distinct antifungal activities against *C. albicans* and *C. glabrata* [31,84]. The antifungal effect of propolis obtained in distinct geographical areas of the European continent was investigated. All propolis used reported an antifungal effect both in reference strains and different species of the *Candida* genus from clinical isolates. Propolis from Ireland and Czechia showed very good fungicidal effects, while propolis from Germany showed mostly fungistatic activity; *C. glabrata*, *C. parapsilosis*, and *C. tropicalis* were the most sensitive *Candida* [28].

In Asia in 2020, Alsayed et al. reported a fungicidal effect of propolis from Saudi Arabia against *C. zeylanoides*, *C. famata*, *C. sphaerica*, *C. guilliermondii*, *C. magnoliae*, and *C. colliculosa* and a fungistatic effect against *C. krusei*, *C. pelliculosa*, and *C. parapsilosis* [85]. In other study, propolis from Turkey showed antifungal activity against different clinical isolates of *Candida* [86]. Similarly, the antifungal effect of aqueous and ethanolic extracts of Iranian propolis was described against *Candida* samples that were collected from 23 oral cavities of patients presenting candidiasis in the oral cavity (isolating 22 samples of *C. albicans* and one *C. glabrata*). Both extracts of Iranian propolis demonstrated inhibitory effects on *Candida*, but the extract that presented greater effectiveness even above the aqueous one was the ethanolic [87]. Other researchers evaluated the antifungal activity of propolis and propolis-loaded nanoparticles (EEP-NPs) from Thailand, observing the impact they have on specific factors that contribute to the pathogenesis of *C. albicans*, where EEP-NPs were mostly active compared to propolis in its free form, inhibiting virulence factors such as adhesion, hyphal germination, biofilm formation, and invasion. It should be noted that the EEP-NPs showed a decrease in the expression of genes related to adhesion processes linked to the hyphae of *C. albicans*, demonstrating that the EEP-NPs have the ability to mediate a great anti-*Candida* activity, attacking key factors of virulence, such as the inhibition of the expression of genes related to adhesion-related proteins, which mediate the morphological change of *C. albicans*, attenuating the virulence of the yeast (Table 8) [88]. In Africa, few studies were found for this review. One was conducted by Papachroni et al. in 2015, who analyzed four propolis of distinct areas of Africa, which showed an effect on *C. albicans*, *C. tropicalis*, and *C. glabrata* [35]. We think that the aforementioned studies that have demonstrated the anti-*Candida* activity of propolis have a significant impact on this issue, since most of them reported fungicidal or fungistatic activity exhibited by propolis from different regions on different strains and clinical isolates of *Candida* [89], as well as some possible mechanisms of action through which propolis inhibits this yeast, such as virulence factors that favor the pathogenicity of *C. albicans*. Some studies even reported the activity of propolis being very promising in in vivo models of candidiasis infections; however, the limitation of all these investigations is that they did not mention the composition of the different propolis, since, as we mentioned earlier, it is important to know and determine the active components present in propolis to identify which molecules are responsible for this antifungal effect [90].

Various studies around the world have focused on the search for components in the propolis that have antifungal effect; below, we describe some of the investigations that revealed the antifungal potential of this natural product. A fraction of the Brazilian red propolis rich in benzophenones was analyzed, which showed activity against different clinical isolates of *C. parapsilosis* and *C. glabrata* resistant to antifungal agents, like fluconazole [91]. Equally, other propolis from Brazil analyzed by different researchers showed fungicide action on different strains, with *C. albicans* being more sensitive and *C. parapsilosis* being the most resistant strain studied. An in vivo study described that gels based on propolis had an antifungal effect similar to clotrimazole cream. In this propolis, different compounds were identified [92]. In Europe, one of the most extensive studies of propolis extracts was conducted, analyzing the effects of 50 different propolis extracts from Polish hives against 89 *Candida* spp. clinical isolates. Most of the samples of propolis produced satisfactory activity, showing high activity in the inhibition of biofilm formation generated by *C. glabrata* and *C. krusei* on the surfaces of polyvinyl chloride and silicone catheters. The propolis inhibited the yeast-to-mycelia morphological change and mycelial growth of *C. albicans*. In addition, the propolis combined with fluconazole or voriconazole on *C. albicans* was shown to have a clear synergism. The chemical composition of three propolis with high and one with low antifungal effect was determined (finding different flavonoids and phenolic compounds), providing evidence that the fungal cell membrane could be the target of propolis [66]. Similarly, in 2019, Pobiega et al. analyzed different Polish propolis (agricultural regions and Southern Poland), the latter being noted by greater antifungal activity against different microorganisms, including *C. albicans* and *C. krusei*, also showing a higher content of bioactive components (Table 9) [43]. Within Africa, samples providing satisfactory results were reported: one of them was Egyptian propolis, which presented antifungal activity against *C. albicans*. In addition, the authors identified compounds such as ferulic acid, cis- and trans-caffeic acids, pinostrobin, and galangine, among others [93]. We agree that the identification of the composition of propolis from distinct areas of the world is crucial, which provides an approach to elucidating some bioactive compounds with antifungal activity and thus paving the way for future research, for example, to complement antifungal drugs with propolis or with its bioactive compounds. However, in the aforementioned studies, the mechanism of action by which propolis or any of its identified molecules exerts their antifungal effect on the different strains of *Candida* must be further investigated [94]. In addition, this pathogen is the cause of many infections in which alternative treatment is needed; propolis could be a promising option.

### 3.2. Trichophyton Infections

The diseases known as dermatophytoses are mycoses generated by fungi that commonly cause different infections in the superficial epithelia in animals and principally in humans. Various filamentous fungi are the cause of these diseases that can invade and acquire nutrients from keratinized epithelia (skin, hair, and nails) [95,96]. Dermatophytoses have the ability to affect people all over the planet, having a higher incidence level in hot tropical countries with high humidity. Approximately 10% to 15% of people are infected by dermatophytes at some time in their life [97]. It is known that dermatophytoses have the ability to affect approximately 25% of the world population according to data from WHO, as well as to generate adults carrying the disease with completely asymptomatic characteristics in a percentage ranging from 30% to 70% [98]. In developed countries, dermatophytes are the main causes of onychomycosis identified with a frequency ranging from 80% to 90%. Around the world, the prevalence of tinea pedis is has been reported approximately at 5.5%, representing 50% of all cases of nail disease [99]. The main cause of these diseases is the genus *Trichophyton*.

The Brazilian Amazon rainforest is a huge source of plant biodiversity, which is why the propolis derived from this area has many biological properties, one of which is the antifungal activity described by Silva et al. in 2015, who stated that red and green propolis are active against strains of *T. rubrum*, *T. tonsurans*, and *T. mentagrophytes*, with red propolis being more efficient than green [100]. Similarly, Brazilian green propolis showed antifungal activity against the preformed biofilms of two clinical isolates of *Trichophyton* from onychomycosis cases, the authors observing that the total biomass and the percentage of living cells of the biofilms that were subjected to the treatment with propolis were lower than in the control for both isolates; therefore, Brazilian propolis had the ability to decrease the number of cells in the preformed *Trichophyton* biofilm. Sixteen patients infected of onychomycosis were treated with topical propolis twice a day, with a 6-month follow-up period. After treatment, the data obtained were encouraging, observing a mycological and clinically total resolution in the nails and showing a complete improvement of the natural morphology of the nail and the disappearance of the fungus of up to 56.25% of patients. Brazilian propolis is a therapy drug with great potential to be used to topically treat onychomycosis caused by *Trichophyton* [101]. Likewise, the Portuguese propolis presented distinct antifungal activity against *T. rubrum* (Table 10) [84]. The different previous studies showed that propolis has variable antifungal activity on different *Trichophyton* strains, inhibits the biofilm of clinical isolates, and a topical treatment based on propolis improved onychomycosis in patients; however, there are several limiting factors in the research of the activities of propolis against this fungus. One of the recurring omissions in this type of research is to omit to description of the components of propolis, since this is transcendental for identifying the bioactive components. Future studies must search for a possible mechanism of action against this pathogen, as it causes very common infections; therefore, it is important for there to be accessible options or traditional medicine to treat them, since a large part of the population uses this type of treatment. In addition, it is vitally relevant to carry out clinical studies that help to validate the distinct doses of propolis that help the treatments, because variety in active principles and the biomedical effects of the several propolis have to be taken into account [102].

### 3.3. Aspergillus and Penicillium Infections

Some species of the *Aspergillus* genus are responsible for chronic pulmonary Aspergillosis (CPA) disease, which can range from nonprogressive to severe effects, such as chronic necrotizing pulmonary Aspergillosis [103]. The number of people around the world who have CPA is estimated at three million, and it is believed that Asia has the highest number of disease cases in comparison to other continents [104]. For the year 2019, it was calculated that after pulmonary tuberculosis, 12 million patients developed CPA [105]. In addition, the species of the genera *Aspergillus* and *Penicillium* produce various secondary metabolites known as mycotoxins [106]. In this set of toxins, we find the aflatoxins, fumonisins, deoxynivalenol, ochratoxin A, and zearalenone are agriculturally important and dietary mycotoxins exposure is associated with many chronic health risks, such as cancer, immune suppression, digestive, blood, and nerve defects [107,108]. Next, studies performed with propolis and these species give these fungal gears are described.

Portuguese propolis presented varying antifungal activity against *Aspergillus fumigatus* [84]. Propolis from different latitudes can present very similar biological activities, as in the case of United States and Chinese propolis against *Penicillium notatum*. With both propolis, the structure and morphology of hyphae were damaged, inhibiting the development of mycelium. Propolis treatment raised extracellular conductivities, showing that propolis probably affects the cell membrane. In addition, a decrease in the activity of enzymes related to the functioning of cellular respiration of *P. notatum* (succinate dehydrogenase and malate dehydrogenase) was observed. Additionally, quantitative proteomic analysis (iTRAQ-based) related to energy metabolism and sterols biosynthetic pathway of *P. notatum* in presence the propolis was described, which showed that 88 proteins (25.8%) were upregulated and 253 (74.2%) were downregulated, of a total of 341 proteins. The major compounds in both propolis were pinocembrin, pinobanksin-3-O-acetate, galanin, chrysin, pinobanksin, and pinobanksin-methyl ether. The authors suggest that all these different properties that propolis has on *P. notatum* can interfere with its development [109]. Similarly, Southern Poland propolis showed antifungal activity against different microorganisms, including *A. niger* and *A. ochraceus*, and were found to contain pinocembrin, chrysin, pinobanksin, apigenin, kaempferol, *p*-coumaric acid, ferulic acid, and caffeic acid [43]. Few studies have been conducted on the effects of propolis against the genus *Aspergillus*, indicating a large gap in the literature. However, the previous studies only focused on observing whether propolis has antifungal activity; only the identified compounds were mentioned, and no correlation was mentioned between the activity and the propolis components, so research is lacking on this topic [110]. We note that study by Xu et al. provides a clear example of how to study natural products on different microorganisms to determine if they present activity or not, to later identify the chemical composition, and then try to find a possible mechanism of action by which natural products could inhibit the pathogen. Finally, the next step in Xu et al.’s research is to conduct studies on in vivo models and clinical trials to provide an alternative and complementary treatment for fungal infections.

Currently, some antifungal drugs are available; however, the problem in treating these diseases is the toxicity toward the host or the emergence of drug resistance in pathogen populations. Here, we describe the efficacy of propolis against different fungal pathogens, where the effect of propolis has been demonstrated in vitro and in vivo [111], as well as in different pathogenicity mechanisms; in some cases, an activity similar to that of the drugs used to treat mycoses has been reported. One of the most remarkable aspects is the use of propolis in clinical studies, where it was shown that propolis can be an alternative that complements the treatment of some of these diseases [112], so it is important to realize more clinical trials to support the effectiveness of propolis in addition to implementing trials focused on evaluating toxicity to determine a standardized dose of propolis that is safe for consumption and application in humans, as well as study the components of each propolis used. These studies could give scientific support to natural products widely used as a therapeutic alternative in rural communities and in developing nations.

## 4. Antiparasitic Activity

Parasitic diseases continue to take an enormous toll on human health globally, particularly in tropical regions [113,114]. Intestinal and protozoan infections are the most common parasitic diseases. Protozoan parasites are unicellular eukaryotes responsible for 1.3 million deaths worldwide annually [114,115]. In several countries, these diseases are unfortunately not a priority with respect to their surveillance, prevention, and treatment. Among these diseases are malaria, Chagas disease, leishmaniasis, trichomoniasis, amebiasis, and giardiasis [116,117,118]. Below, we discuss the studies that have been conducted with propolis on the pathogens that cause these diseases.

### 4.1. Malaria

Malaria is a disease that can cause death generated by a protozoan parasite of the Plasmodium genus. This disease is transmitted by female Anopheles mosquito bites. It is estimated to impair about 219 million people each year in 87 countries, mainly affecting pregnant women and children aged between 0 and 5 years [119]. Propolis has been used from some countries to study its antimalarial effects on species of the genus Plasmodium. In an effort to find alternatives for this disease, 20 propolis from different provinces in Cuba were evaluated in vitro, with three showing significant activity on *Plasmodium falciparum*. Chemical composition analyzes were carried out for propolis, where compounds of phenolic origin and triterpenes such as linquiritigenin and lupeol were found; these compounds were already reported to have activity against *P. falciparum* [120]. Similarly, twelve propolis from Libya were evaluated in vitro and demonstrated antiprotozoal activity, including against *P. falciparum* [121]. Propolis also showed antimalarial properties in vitro, for example, in the case of four Iranian propolis that showed in vitro and in vivo activity at different concentrations against *P. falciparum*. The chemical composition of the two extracts with higher activity was determined, and molecules such as palmitic acid, stearic acid, pinocembrin, tectochrysin, and 4′,5-dihydroxy-7-methoxyflavanone with an antiplasmodial effect were identified [122]. Saudi propolis considerably suppressed parasitemia and demonstrated an important effect on decreasing anemia in *Plasmodium*
*chabaudi*-infected mice, reducing oxidative damage by enhancing the catalase function and the glutathione concentrations, and enhancing the quantities of pro-inflammatory cytokines. It is reported that these cytokines promote phagocytosis, chemotaxis, and antibody-dependent cytotoxicity. Furthermore, they are responsible for the activation of neutrophils as well as protection against this parasite (Table 11) [123]. These works demonstrate the promise of using propolis against this complex disease; however, some of these works lacked a chemical analysis of propolis. As mentioned previously, it is necessary that future studies with propolis include the origin and description of the chemical composition [6]. Some of the compounds such as lupeol and liquiritigenin have been reported to have antiplasmodial activity [124,125] and could be related to the effects of Cuba’s propolis against this parasite. However, it is still necessary to realize different studies to better understand the effects of propolis in the treatment of this disease.

### 4.2. Chagas Disease

*Trypanosoma cruzi* is a protozoan parasite responsible for Chagas disease, which is transmitted mainly by Hematophagous triatomine insects, and to lesser extent by oral, congenital, blood transfusion and organ transplantation [126]. This disease is a problem for most of Latin America and especially affects marginalized zones; it is estimated that 8–10 million people are infected each year [126,127]. Extensive research has been conducted on propolis as an anti-trypanosome agent using in vitro models. We highlight the thorough study on various propolis from Brazil, of which the activity of propolis extract on *T. cruzi* trypomastigotes was reported [30]. Some propolis were effective on the three forms of the parasite, and treatment with propolis strongly inhibited infection levels by promoting lysis of bloodstream trypomastigotes and diminished the number of parasites in peritoneal macrophages and infected heart muscle cells [128]. Some propolis from Brazil showed an in vitro effect against *T. cruzi*; their chemical composition was determined, and caffeic acid, cinnamic acid, pentenoic acid, ferulic acid, linoleic acid, amyrin, and pinostrobin, amongst others, were identified; however, in these studies, the anti-trypanosomal activity of these compounds was not evaluated [129,130]. Interestingly, other authors reported that the application of natural products obtained from propolis produced anti-trypanosome effects; for example, four components were isolated from Brazilian propolis and two were effective on *T. cruzi* [131]. Similarly, Brazilian and Bulgarian propolis were shown to have activity against this parasite, diminishing replication of the parasite without damaging the membrane of the host cell. Microscopic analysis showed that the main organelles damaged by the extracts were mitochondrion and reservosomes [132]; two Bulgarian propolis share many bioactive compounds, mainly flavonoids and a remarkable antitrypanosomal effect; epimastigotes were more sensitive than trypomastigotes. The efficacy of either of the two Bulgarian propolis on trypomastigotes was similar to that of the reference drug [133].

Despite the encouraging results from the in vitro tests, in vivo studies are scarce. For example, treatment with Bulgarian propolis in *T. cruzi*-infected mice led to a reduction in parasitemia and showed no toxic hepatic or renal effect, the spleen mass decreased, and the initial inflammatory reaction was modulated, favoring a greater number of CD8^+^ and partially inhibiting the increase in CD4 [134]. Studies conducted with propolis from Brazil in infected mice recorded a decrease in the number of parasites and mortality of the animals without generating toxicity or injury on other tissues, so it could be assayed in combination with other drugs as a potential metacyclogenesis blocker (Table 12) [135].

### 4.3. Leishmaniasis

Leishmaniasis is a neglected disease group, occasioned by 20 species of protozoan parasites belonging to the genus *Leishmania* and spread by female sand flies of the genus Phlebotomus or Lutzomyia. Present in nearly 100 countries and endemic in Asia, Africa, the Americas, and the Mediterranean region, more than 12 million people, about 25,000 deaths, and 1 million new cases are reported annually; according to the WHO, it is a Category I (emerging or uncontrolled) disease [136]. In humans, four clinical forms of this disease can develop: visceral leishmaniasis (VL), cutaneous leishmaniasis (CL), mucocutaneous leishmaniasis (MCL), or post-Kala-azar dermal leishmaniasis (PKDL). For more than six decades, pentavalent antimonials (SbV) were the first-line drugs against leishmaniasis; however, the toxicity and resistance of the parasites are the main limitation of these drugs [137]. Other treatments such as pentamidine, paromomycin, or amphotericin B has been employed, but its high costs and side effects make it difficult to use [137,138]. Therefore, alternatives are urgently required that complement and help with the adequate treatment of leishmaniasis. Propolis has been studied as an alternative to various protozoa, including parasites of the genus *Leishmania*. The main investigations on the leishmanicidal effect of propolis have mainly been conducted in vitro, focusing on determining the effect of propolis on the mortality of this parasite, e.g., the brown, green, and red propolis from Brazil and that from Portugal, which showed significant growth inhibition of *L. braziliensis* Vianna, *L. infantum*, and *L. amazonensis* promastigotes, and decreased the number of internalized amastigotes in infected murine macrophages [84,139,140,141]. Some of the countries with the highest incidence of Leishmaniasis are in the Middle East; the administration of propolis as an antileishmania agent has also been reported in this region. The composition of propolis from three regions of Turkey (Adana, Hatay, and Bursa) was analyzed, and differences were found in the type of compounds and in their quantities; the main component of Adana propolis was found to be cembrene, that of Hatay was chrysin, and Bursa’s was cinnamyl cinnamate. All three registered a good antileishmanial activity against *L. tropica* or *L. infantum*, but the propolis from Bursa was the most effective [142,143]. The effects of propolis on Leishmaniasis have been studied more: as mentioned earlier, various in vitro studies have shown the benefits of this apiculture product. Fortunately, in several of these studies, the chemical compounds present in each propolis were identified. Several of these pure compounds have been tried individually against different species of *Leishmania*, and these compounds are likely related to the antileishmanial effect of propolis [144,145,146]. Although these studies are limited, since they only involved in vitro tests, they provide support for the use and application of propolis in animal models.

Many have evaluated the effect of propolis from Latin American countries on various species of the genus *Leishmania* and have identified a large part of the chemical composition of these propolis. For example, 20 propolis from Cuba presented in vitro antimicrobial properties; the major effect was found on *L. infantum*. The results demonstrated an association between the biological effect and compounds identified. The propolis that contain acetyl triterpenes as amyrin, lupeol, and cycloartenol as the main constituents are the best options for future studies [120]. Similarly, three propolis obtained from distinct areas in Ecuador (Quito, Guayaquil, and Cotacach) avoided *L. amazonensis* growth, highlighting the activity of sample rich in flavonoids as naringenin, sakuranetin, eupatolitin, and rhamnazin [147]. Brazil is one of the countries with the most studies of the biological and chemical properties of its propolis. Brazilian propolis (Ribeirao Petro and Minas Gerais) were proven on *Leishmania* species associated with different clinical forms of leishmaniasis, and the chemical composition was determined. Propolis from Minas Gerais showed great antileishmanial effect on *L. amazonensis*, *L. braziliensis*, *L. chagasi*, and *Leishmania major*, with the last species being the most susceptible. Ribeirao Petro propolis was only evaluated against *L. amazonensis*; it recorded a dose-dependent activity against promastigotes, also the number of parasites decreased inside macrophages. Although a leishmanicidal effect on *L. amazonensis* was reported in the two studies, the effects were not the same, because the extracts from Minas Gerais and Ribeirao Petro had different chemical compositions: the main compounds of the first were diethyl 2-methylsuccinate, cinnamic acid, pentanedioc acid, and hydrocinnamic acid; for the second, they were artepillin C, 4,5-dicaffeoylquinic acid, *p*-coumaric acid, and drupanin [148,149]. Brazilian propolis also showed strong in vivo effects: in an experimental infection model with *L. braziliensis* using BALB/c mice treated previusly with propolis, it reduced growth and promoted morphologic alterations on promastigotes and also favored the TNF-α levels in supernatants from liver cells and peritoneal exudate [150]. Green propolis decreased more than 75% in lesion development caused by L. brazilienzis, while the glucantime treatment showed a 57.7% decrease (Table 13) [151].

When used in combination with nitric oxide (NO) during infection with *L. amazonensis* at the lesion site, the levels of NO, healing, collagen synthesis, the function of macrophages and fibroblasts were favored, in addition to decreased parasitized cells, pro-inflammatory factors, and tissue damage [152]. Propolis from Brazil was also used in combination with first line antileishmaniasis medications. Green propolis was administered in combination with liposomal meglumine antimoniate, decreasing the parasitic burden in the liver without damaging or altering the functions of the kidney, liver, spleen, and heart (Table 14) [153]. The two works mentioned in this section are noteworthy as they used propolis as a complementary or combination treatment with another substance. This type of study demonstrates the path that can be followed in the examination of propolis and its bioactive compounds, since propolis is not intended to replace existing treatments but to supplement them with new alternatives [154,155]. Finally, clinical trials are still needed to demonstrate the effectiveness of propolis in humans.

### 4.4. Giardiasis

Giardiasis is a parasitic intestinal disease, the etiological agent of which is *Giardia duodenalis*, also known as *G. intestinalis* or *G. lamblia*. This parasite is transmitted mainly by consuming water or food that contains Giardia cysts. Symptoms of infection are usually diarrhea, nausea, epigastric pain, and weight loss. Giardiasis annually affects about 200 million people worldwide. Since 2004, it is listed by the WHO as a neglected disease by the World Health Organization. The prevalence of *Giardia* infection is higher in developing countries [156,157]. Several remedies of traditional medicine have been administered as a complement to treat this disease [158], among which is propolis.

There are reports of the in vitro effect of three samples from Sonoran Desert propolis in Mexico (Caborca, Pueblo de Alamos, and Ures) and some of its bioactive compounds. The Ures propolis presented a remarkable activity on *G. lamblia* in a dose-dependent manner, as well as one of its components (CAPE), which registered the highest antigiardia effect [159]. Similarly, propolis from Brazil showed efficacy in eliminating trophozoites of *G. lamblia*. The effect on the proteolytic activity of excretory/secretory products (ESPs) from trophozoites treated with propolis was studied; however, no significant differences were found between hydrolysis patterns and inhibition on the protease activity of propolis-treated and untreated trophozoites [160,161]. Propolis from Egypt was reported to be effective in an in vivo model of giardiasis with immunodeficient mice. The propolis produced a diminution in intensity of infection, as well an augment in the IFN-γ serum level and in the CD4^+^:CD8^+^ T cell ratio. Combination propolis and metronidazole presented a great effect in reducing the number of parasites than that produced by each drug alone. Futhermore, this combination induced an immunological regulation, mainly in T lymphocytes, which favor intestinal homeostasis and histological integrity (Table 15) [162]. While several drugs against this parasitosis are available, its incidence continues to be higher in developing countries [163]. In these regions, it is common for people to use traditional or alternative medicine to address health problems [164]. For this reason, it is necessary to determine the effectiveness of these treatments against this parasite. The works included in this review on this disease demonstrate the in vitro and in vivo effect of propolis and its effect on the immunological response against *G. lamblia*. Further research is important to support the utility of propolis as an alternative against Giardiasis; the work conducted to date and the information available are limited.

In this work, we addressed the properties of propolis in the defense against these parasitic diseases; in most cases, propolis showed interesting and promising antiprotozoal activities [165]. However, several challenges remain, such as the wide variety in the components of propolis in each geographical area, isolating the components responsible for the activities, and describing their mechanisms of action and synergy. It is also necessary to promote and conduct research using animal models, since very few studies have been published. It should be noted that the implementation of clinical studies is necessary to support the antiparasitic activity of propolis, as well as conducting research focused on the combined treatment of propolis with different drugs used to treat parasitic infections in humans, and to find more effective complementary treatments in order to be able to reduce the dose and toxicity of the drugs currently implemented.

### 4.5. Helminths

Humans are exposed to a remarkable number of parasites, including protozoans (over 70 species), helminths (about 300 species), and arthropod parasites. There are two major phyla of helminths: the nematodes (also known as roundworms) and the Platyhelminthes (Trematoda and Cestoda) [166,167]. According to the WHO, more than 2 billion cases of intestinal worms were registered in 2018, mainly affecting disadvantaged communities [168]. However, this number could be higher, since many of these diseases are not reported [169]. Another relevant aspect regarding helminths is that they not only parasitize humans, but also affect many domestic animals and the livestock industry, resulting in large economic losses [168]. In addition, in the absence of therapeutic options in developing regions, this population resorts to the use of traditional remedies or alternative medicine to treat these parasites.

The anthelmintic activity of some propolis from Egypt on adult flukes of *Fasciola gigantica* was reported. Alteration of the architecture was found as lifting base of the spines and large blisters in the apical cone, several of which seemed to have burst, generating injuries. The inhibitory activity on the viability and hatchability of immature *F. gigantica* eggs was also found, showing the highest inhibitory effect compared with other treatments. The chemical composition of these propolis was determined. Compounds such as diprenyl-dihydrocoumaric acids, coumarate esters, ferulate esters, hydroxy acetophenones, furanon derivative, furofuran lignans, benzofuran lignans, and valeric acids derivatives could be related to anthelmintic activity [170,171]. Other propolis from Egypt was evaluated against *Schistosoma mansoni* in mice: propolis alone or in combination with praziquantel were administrated. Propolis administration did not eliminate the worms of infected mice but significantly reduced the hepatic granuloma number, hepatic, splenic, and plasma myeloperoxidase (MPO) activity, as well the liver and thymus NO levels, and also regulation of plasma antioxidant proteins evidenced by decrease in malondialdehyde (MDA) and normalization of glutathione (GSH) [172]. The antihelmintic activity of propolis from Turkey on *Echinococcus granulosus* was reported: 1 µg/mL of propolis killed all the protoscoleces in the in vivo part of the study, without causing side effects when administered intraperitoneally. However, the mechanism of action and chemical composition were not reported [173]. There are also reports of the anthelmintic effect of the essential oil of Brazilian red propolis, larvae of *Toxocara cati*, were incubated during 48 h with the essential oil, and then later inoculated in mice. The authors informed 100% effectivity to disable the infective capacity of the larvae [174]. Five propolis from distinct parts of Libya were studied, and they presented moderate activity against *Trichinella spiralis*. The components of the propolis was analyzed, and fourteen compounds were identified, of which cycloartanol, mangiferolic acid, agathadiol, isocupressic acid, and isoagatholal were highlighted (Table 16) [175]. These compounds may play interesting roles in the effects of propolis on helminths in each of the studies reviewed. As each propolis has a complex and changing chemical composition, it is a priority to determine whether the antihelmintic activity is due to a specific compound or a synergism phenomenon to identify new pharmacological alternatives [176]. Since the results in these works show the favorable effects of propolis against various helminths, these propolis could be tested against parasites such as *Tenia*, *Enterobius*, and *Ascaris*, which have high incidence in several countries and are a public health problem [177,178].

Helminth parasites mainly continue to be a public health problem in countries with disadvantaged and low-resource communities. Currently available anthelmintic drugs include the benzimidazoles (albendazole and mebendazole), pyrantel pamoate, and ivermectin [179]. Although these drugs are usually well-tolerated and efficient for the treatment of helminth parasites, they are limited in number, and the susceptibility among helminth species has been shown to vary greatly in different populations [180]. Another concern is the emergence of resistance, which have mainly been observed in veterinary medicine over the past decade [181]. Traditional and alternative medicine offers a wide repertoire of compounds that could complement the treatment of these diseases. Research in animal models must be increased, and clinical trials are needed to confirm the safe utilization of propolis in these diseases.

## 5. Antiviral Activity of Propolis

Viruses need the host cells’ biosynthetic machinery to replicate [182,183]. Viral infections are responsible for some diseases in humans and cause serious public health problems in populations worldwide [184]. Therefore, recent research on new antiviral medications is increasing due to the development of resistance to antiviral drugs [185]. As such, the study of natural products that present antiviral activity, such as propolis and some of its identified compounds, is vital [186,187].

Two propolis from Czech Republic (aqueous and ethanolic extracts) were studied, and both showed great antiviral effect on herpes simplex virus type 2 (HSV-2). Both propolis decreased the infection and exhibited a concentration- and time-dependent antiviral effect. Additionally, the two propolis showed a high antiviral effect when viruses were pretreated prior to infection; thus, both propolis could be used to treat recurrent herpetic infection topically [188].

In other research, the antiviral effect of different propolis from several geographic regions, such as the United States, Brazil, and China, was against human immunodeficiency virus type 1 (HIV-1). All propolis inhibited viral expression in CD4^+^ lymphocytes and microglial cell in a concentration-dependent manner. In another study, propolis from the United States suppressed cell fusion HIV-1 in cultures of CD4^+^ lymphocytes, suggesting that the possible mechanism of propolis’ antiviral property in CD4^+^ lymphocytes is produced in part by inhibition viral entry into cells [189].

The antiviral activity of four Brazilian propolis on influenza virus was examined, and all propolis presented anti-influenza virus effect in vitro. In this same study, the four propolis were studied in a murine influenza virus infection model (propolis was orally administered three times daily for seven days), and only one propolis sample effectively prolonged the lifetime of infected mice. The authors concluded that the Brazilian propolis possessed antiviral effect and amelioration influenza symptoms in mice [190]. In another study, three different extracts of Brazilian propolis were administered orally three times daily for six days to cutaneous herpes simplex virus type 1 (HSV-1)-infected mice to study their effect on HSV-1 infection. The three propolis presented anti-HSV-1 activity and favored immunological effect on intradermal HSV-1 infection in mice (Table 17) [191]. At present, diseases caused by viruses are a priority in any health system due to the severity of their symptomatology, their high infective capacity, and mortality. Viral diseases can affect the economy of a country or the globe [192,193]. The previously mentioned propolis studies are notable and highly relevant, since they reported antiviral properties capable of inhibiting viral replication, cell fusion in cultures of CD4^+^ lymphocytes, and stimulation of immunological activity [189]. However, these works lack a chemical analysis, which limits the explanation of some possible action mechanisms related to the secondary metabolites present in each propolis.

Some flavonoids and phenolic acids, also described in propolis, presented antiviral activity [187]. One antiviral study showed that Canadian propolis had a pronounced viricidal effect against HSV-1 and HSV-2 and interfered with virus adsorption. Different compounds were identified in this propolis. The interaction with propolis indicates damage to the HSV and suggests that propolis could damage protein components of envelopes essential for adsorption and penetration of the virus into the cells (Table 18) [194].

Similarly, the antiviral effect of aqueous and ethanolic extract of Czech Republic and some compounds identified in propolis against HSV-1 in cell culture was analyzed. Both samples presented high anti-HSV-1 effect in cells effect in prior to viral infection; of the compounds tested, galangin and chrysin were the most active components. However, the propolis with various compounds presented higher antiviral activities than the isolated constituents alone. The authors concluded that the antiherpetic activity of propolis is due to a combination of several components; therefore, the propolis from Czech Republic was found to be more effective on herpes infection than the individual compounds [187].

In another study, the replication of HSV-1 and HSV-2 was inhibited with the propolis from the south of Turkey. Propolis started to suppress HSV-1 replication after 24 h of incubation and effect on HSV-2 started at 48 h after incubation. This activity of propolis on HSV-1 and HSV-2 was checked by a lower in the number of viral copies. They found that propolis showed activity similar to that of acyclovir, since both started to suppress HSV-1 replication following 24 h of incubation. They also found a synergistic effect of combined propolis and acyclovir on HSV-1 and HSV-2 replication compared with acyclovir alone. Some compounds in the propolis were identified. The propolis from the south of Turkey was found to present relevant antiherpetic activities in comparation with acyclovir; particularly, the synergism generated by the antiherpetic effect of propolis and acyclovir in combination has a stronger activity on HSV-1 and HSV-2 than acyclovir alone. The authors mentioned that the possible mechanism of synergism between acyclovir and propolis may be attributed to some of the components of propolis [185]. As described in each work, most of the molecules identified in each propolis are compounds of phenolic origin. Some of these have been reported as having the ability to stimulate antiviral responses in in vitro and in vivo models, promoting the production of interferons as well as the activation of cytotoxic T lymphocytes and natural killer cells [195,196,197,198,199]. This supports the use of propolis as a source of new molecules with antiviral effects and an alternative to complement the treatment of these diseases.

Although little literature about the antiviral activity of propolis in clinical studies exists, in 2019, Jautová et al. reported that one lip cream with propolis extract from Central Europe produced a better effect than acyclovir to treat patients with herpes labialis in the vesicular phase, confirming the clinical efficacy of lip cream composed of European propolis in the early and late start of treatment during an episode of herpes labialis [200]. Similarly, a clinical study reported the contribution of propolis extract from Central Europe as a constituent in a lotion for complementary treatment of Herpes zoster. A total of 33 patients with a diagnosis of Herpes zoster applied a treatment with a propolis-based lotion for 28 days as a complementary treatment to oral antiviral treatment with acyclovir. The healing of lesions was improved and faster with the propolis treatment; approximately 50% of propolis-treated patients had no injuries on day 14 and the growth of new vesicles was inhibited, clinically confirming the antiviral effects of European propolis and demonstrating the properties of complementary therapy on the systemic antiviral treatment of Herpes zoster (Table 19) [201].

The current situation related to COVID, which has compromised all health systems, makes it necessary to search for therapies that prevent or mitigate the complications of this disease. Natural products such as propolis are an interesting option in the search of complete therapies. Some recent research mentions the potential benefits of using propolis against this disease. These studies are based on previously reported activities against other viruses and on in silico models that allow predictions of activities against this virus. These studies focus mainly on reported bioactive compounds in the different propolis; they include antiviral activities that could be applied against SARS-CoV-2 or immunomodulatory effects that would reduce the symptoms of the disease. One of the clearest examples is quercetin, one of the most abundant and consumed flavonoids in the diet. Quercetin has been shown to inhibit the replication cycle of the virus, since it reduces the functioning of the main protease (Mpro) and S protein of SARS-CoV-2. CAPE, one of the main components of many propolis, is able to inhibit the transmembrane protease serine 2 (TMPRSS2), angiotensin-converting enzyme-related carboxypeptidase (ACE-II), and Mpro; these molecules are crucial for access and replication viral of SARS-CoV-2 in cells. Another interesting compound is the rutin that reduced the function of S protein, ACE-II, and others non-structural proteins of SARS-CoV-2. These flavonoids are also able to regulate JAK/STAT-mediated signaling and the production of ROS, NO, pro- and anti-inflammatory cytokines, avoiding a cytokine storm. They even reduce the risk of comorbidities that complicate the betterment of patients with COVID-19 [202,203,204]. Propolis and its bioactive compounds open new means for future works that describe in detail their effects on SARS-Cov-2 and are able to be applied as a complementary therapy in clinical studies.

Considering all the research mentioned above, the search for new strategies for the control and complementary treatments of infections caused by viruses has become a global public health priority. However, more in vitro and clinical studies with propolis are needed to elucidate its mechanisms of action and identify the molecules responsible for the antiviral effects of this natural product.

## 6. Conclusions

We collect the main studies of the effect of propolis on pathogens related to infectious diseases of medical relevance. The reports of the efficacy of the different propolis are encouraging: this bee product showed effectiveness on bacteria, fungi, protozoa, helminths, and viruses. Propolis presents a great spectrum of components that could be used to treat characteristic affections of distinct diseases. Not all propolis present the same activities; depending on the flora of the geographical area, each propolis has a different chemical composition with unique biological activities, making propolis a promising source of discovering molecules, which can be used in different clinical situations. Propolis offers potential for research into the treatment of infectious diseases that lack adequate therapies due to the resistance of pathogens to drugs, either isolating active components to be studied alone or combined with different current drugs. Despite the in vitro and in vivo evidence suggesting that propolis can be a reliable alternative to existing drugs, the effect of propolis must be investigated in the clinic to improve tour comprehension of the mechanisms of action of the different propolis, attain the synergism of their compounds, and generate a standardized and safe consumption protocol.

Another relevant aspect is that clinical tests with propolis, bee products, or other natural products are scarce but necessary. From products used in traditional medicine, modern medicine has obtained compounds such as taxol, valproic acid, polycarpine, ephedrine, digoxin, and acetylsalicylic acid, just to name a few. The therapeutic uses and applications of natural products and their derivatives are promising in the search for new treatments, so clinical studies against diseases caused by microbes resistant to drugs or treated with toxic agents should be a priority in future clinical research.

Finally, a new perspective to consider in future research is to investigate the presence and function of microRNAs (miRNAs) in propolis. Recent studies have proposed that the miRNAs present in honey from plants visited by bees during their collection could play a determining role in the development of larvae. The finding of these molecules could be surprising related to the beneficial effects on the health of consumers of this bee product. The identification of miRNAs in propolis would be crucial to understanding and explaining many of its biological and medicinal activities, and these activities are currently attributed mainly to compounds such as flavonoids and terpenes. miRNAs in bee products can be the subject of various investigations, and their clinical applications could generate new treatments based on nutritional supplements with various specific benefits for health [205,206].

## Figures and Tables

**Table 1 biology-10-00428-t001:** Effect of propolis from several parts of the world on various *Staphylococcus* species.

Propolis Origin	Bacterial Species	Activity	Ref.
Brazil (red, green, and brown propolis from different regions)	*S. aureus*	Red extracts showed highest activity compared with green and brown extracts (MIC: 25–200 μg/mL both propolis)	[30]
Poland	Twelve MSSA and MRSA clinical isolates	Displayed variable effectiveness against twelve clinical isolates (MBC: 0.78–3.13 mg/mL)	[32]
Germany, Ireland, and Czech Republic	*S. aureus*, MRSA, and *S. epidermidis*	The three propolis showed moderate antibacterial activity (MIC: 0.08–2.5 mg/mL)	[28]
Italy	*Staphylococcus* spp. and *S. aureus*	Propolis (MIC: 0.31–2.5 mg/mL) showed inhibitory action on the lipase activity of 18 *Staphylococcus* spp. and an inhibition on coagulase of 11 *S. aureus* strains; showed inhibition of adhesion and consequent biofilm growth of *S. aureus*	[33]
Serbia (53 samples from different regions; blue and orange propolis)	*S. aureus*	All propolis samples showed antibacterial activity, with orange-type (0.1–14.7 mg/mL) showing higher activity than blue-type propolis samples (1.8–12.9 mg/mL)	[34]

MIC: minimum inhibitory concentration; MBC: minimum bactericidal concentration.

**Table 2 biology-10-00428-t002:** Antibacterial effect of diverse propolis and its chemical composition.

Propolis/ Compound	Bacterial Species	Identified Compounds	Activity	Ref.
Brazil (red, green, and brown propolis from diverse regions)	*S. aureus*	Ferulic acid, gallic acid, caffeic acid, coumaric acid, *p*-coumaric acid, catechin, drupanin, kaempferide, artepillin C, luteolin, and pinocembrin	All propolis showed distinct antibacterial activities (200–1600 µg/mL)	[40,41,42]
Poland (25 different samples)	*S. aureus* and *S. epidermidis*; 16 MSSA and four MRSA clinical isolates	Flavonols, flavones, flavanones, pinocembrin, chrysin, pinobanksin, apigenin, kaempferol, *p*-coumaric acid, ferulic acid, and caffeic acid	All propolis (1–8 mg/mL) showed distinct antibacterial effect on *S. aureus* and *S. epidermidis*; in the clinical isolates, all the samples of propolis presented different activities and two of them showed higher antistaphylococcal activity; propolis effectively eradicated staphylococcal biofilm	[27,43]
Pinocembrin, galangin, and chrysin (South African propolis)	*S. aureus*	Pinocembrin, galangin, and chrysin	The combinations of these three flavonoids presented higher inhibition than alone flavonoids (0.04–0.26 mg/mL)	[44]

**Table 3 biology-10-00428-t003:** Effects of propolis in combination with various drugs on different bacterial species.

Propolis Origin	Antibiotics	Bacterial Species	Activity	Ref.
Ireland	Two-drug combinations: vancomycin, oxacillin, and levofloxacin	MRSA	Propolis (MIC: 0.4–5 mg/mL) synergistically enhanced the efficacy of antibiotics, especially those acting on cell wall synthesis (vancomycin (0.2 mg/mL) and oxacillin (12.5 mg/mL)) on drug-resistant bacteria	[28]
Poland	Amikacin, kanamycin, gentamycin, tetracycline, and fusidic acid	*S. aureus*	Propolis (16–32 µg/mL) showed a synergistic effect in combination with various antibiotics (1–0.0312 µg/mL) that inhibit protein synthesis	[27]
	Cefoxitin, clindamycin, tetracycline, tobramycin, linezolid, trimethoprim + sulfamethoxazole, penicillin, and erythromycin	*S. aureus* clinical isolates	The combination of propolis (MIC: 0.39–0.78 mg/mL) with different drugs potentiated the antibacterial effect of eight antistaphylococcals (1–30 µg/mL) against all strains	[32]
Italy	Ampicillin, gentamycin, streptomycin, chloramphenicol, ceftriaxone, and vancomycin	*S. aureus* and *S. epidermidis.*	Propolis increased the antibacterial effect of ampicillin (0.05–3.12 µg/mL), gentamycin (0.05–1.56 µg/mL), and streptomycin (0.05–50 µg/mL); moderately for chloramphenicol (0.05–25 µg/mL), ceftriaxone, and vancomycin (0.39–3.15 µg/mL)	[33]
Nanoparticles prepared with Malaysian propolis	Rifampicin, ciprofloxacin, vancomycin, and doxycycline	*S. epidermidis*	Propolis nanoparticles (15.63–125 µg/mL disrupted bacterial biofilms by causing membrane damage and significantly reducing biofilm formation, and showed synergism with antibiotics (0.2–25 µg/mL)	[46]

MIC: minimum inhibitory concentration.

**Table 4 biology-10-00428-t004:** Antistreptococcal activity of diverse propolis and its chemical composition.

Propolis/ Compound	Bacterial Species	Identified Compounds	Activity	Ref.
Germany, Ireland, and Czech Republic	*S. pyogenes* and *S. pneumoniae*	N. I.	Moderate antibacterial activity (MIC: 0.6–5 mg/mL)	[28]
Mexico	*S. mutans*	Pinocembrin, chrysin, galangin, alpinetin, dillenetin, isorhamnetin, ferulic acid, syringic acid, and caffeic acid	Propolis (MIC: 125–1024 µg/mL) presented antibacterial activity; galangin, ferulic acid, syringic acid, and caffeic acid showed activity against this oral pathogen	[50]
South of Brazil (different samples)	*S. mutans*	Gallic acid, caffeic acid, coumaric acid, artepillin C, and pinocembrin	All samples of propolis (25–800 µg/mL) have an inhibitory action biofilm growth	[42]
Iran/ quercetin	*S. mutans* and *S. pneumoniae*	Quercetin	Both propolis (MIC: 3.12–100 µg/mL) were efficient against the bacteria studied and showed an inhibitory activity *S. mutans* biofilm adherence	[51]

N.I., none identified; MIC: minimum inhibitory concentration.

**Table 5 biology-10-00428-t005:** Antibacterial activity of different propolis on various microorganism species.

Propolis/ Compound	Bacterial Species	Identified Compounds	Activity	Ref.
Germany, Ireland, and Czech Republic	*E. coli*, *S. choleraesuis*, and *S. flexneri*	N.I.	All showed moderate antibacterial activity (MIC: 0.6–5 mg/mL)	[28]
Brazil (red, green, and brown propolis) and Southern Poland	*E. coli* and *L. monocytogenes*	Ferulic acid, *p*-coumaric acid, caffeic acid, catechin, luteolin, drupanin, kaempferide, artepillin C, pinocembrin, chrysin, pinobanksin, apigenin, and kaempferol	All samples showed distinct antibacterial activities (25–800 µg/mL)	[40,41,43]
Pinocembrin, galangin, and chrysin (South African propolis)	*L. monocytogenes* and *E. coli*	Pinocembrin, galangin, and chrysin	The combinations of these three flavonoids (0.04–0.26 mg/mL) presented higher inhibition activity than alone components	[44]

N.I., none identified; MIC: minimum inhibitory concentration.

**Table 6 biology-10-00428-t006:** Antibacterial activity of diverse propolis on different microorganisms that cause nosocomial infections.

Propolis Origin	Bacterial Species	Activity	Ref.
Brazil (red, green, and brown propolis from different regions)	*Klebsiella* sp.	Red extracts showed higher activity than green and brown extracts (MIC: 31.1–1000 µg/mL both propolis)	[30]
Germany, Ireland, and Czech Republic	*P. aeruginosa*, *H. influenzae*, *K. pneumoniae*, and two clinical isolates of *K. pneumoniae*	All propolis showed moderate antibacterial activity (0.06–2.5 µg/mL)	[28]
Cameroon, Congo, and Kenya	*K. pneumoniae* and *P. aeruginosa*	All propolis showed differences in antibacterial activity (50 mg/mL)	[35,36]

MIC: minimum inhibitory concentration.

**Table 7 biology-10-00428-t007:** Effect of diverse propolis and its main identified components on *P. aeruginosa* and *K. pneumoniae*.

Propolis/ Compound	Bacterial Species	Identified Compounds	Activity	Ref.
Albania	*P. aeruginosa*	Caffeic acid, *p*-coumaric acid, ferulic acid, isoferulic acid, quercetin, apigenin, pinobanksin, chrysin, pinocembrin, galangin, and CAPE	Propolis (15.6–62.5 mg/mL) inhibited the microbial development and biofilm growth, also decreased extracellular DNA release and phenazine production	[61]
Pinocembrin, galangin, and chrysin (South African propolis)	*P. aeruginosa* and *K. pneumoniae*	Pinocembrin, galangin, and chrysin	Combinations of the three flavonoids (0.6–25 µg/mL) present better antibacterial effect than single components	[44]

**Table 8 biology-10-00428-t008:** Anti-*Candida* activity of different propolis around the world.

Propolis Origin	Fungal Species	Activity	Ref.
Brazil (green propolis)	Vaginal isolates of *C. albicans*	Suppress growth and biofilm formation	[81]
Brazil	*C. albicans*, *C. glabrata*, *C. tropicalis*, *C. guilliermondii*, and *C. parapsilosis*	All strains were suppressed, with minimal variation independent of the yeast species (273.43 μg/mL)	[80]
	*C. albicans*	Showed fungicidal activity against the three morphogenetic types; the induced cell death was mediated by metacaspase and Ras signaling	[82]
Brazil (topical pharmaceutical preparation based upon propolis)	Vulvovaginal candidiasis infection in a mouse model (*C. albicans*)	Can partially control *C. albicans* infections (0.05%, 0.1%, and 0.2%)	
Poland (different samples)	Azole-resistant *C. albicans*, *C. glabrata*, and *C. krusei* clinical isolates	Only one of the four propolis samples revealed high activity (MFC: 0.0006–1.25% *v*/*v*)	[83]
Portugal and France	*C. albicans* and *C. glabrata*	Presented distinct antifungal activities (15.63–250 μg/mL)	[31,84]
Germany, Ireland, and Czech Republic	*C. albicans*, *C. glabrata*, *C. parapsilosis*, *C. tropicalis*, and *C. krusei*	Propolis from Ireland and Czech demonstrated excellent fungicidal (0.1–5 mg/mL) effects; propolis from Germany showed mostly fungistatic (0.1–2.5 mg/mL) activity. *C. glabrata*, *C. parapsilosis*, and *C. tropicalis* were the most sensitive.	[28]
Saudi Arabia	*C. zeylanoides*, *C. famata*, *C. sphaerica*, *C. guilliermondii*, *C. magnoliae*, *C. colliculosa*, *C. krusei*, *C. pelliculosa*, and *C. parapsilosis*	Showed fungicidal (2.5% *v*/*v*) and fungistatic (5%) effects against different strains	[85]
Iran	22 samples of *C. albicans* and one sample of *C. glabrata* isolates from oral cavities of patients with clinical oral candidiasis	Both extracts showed inhibitory effects on *Candida*, but the extract that presented greater effectiveness even above the aqueous (0.2–130 mg/mL) one was the ethanolic (0.4–210 mg/mL)	[87]
Propolis-loaded nanoparticles from Thailand	*C. albicans*	Inhibited the virulence factors of *C. albicans*, such as adhesion, hyphal germination, biofilm formation, and invasion (1 and 2 mg/mL)	[88]

MFC: medium fungicidal concentration.

**Table 9 biology-10-00428-t009:** Antifungal activity of diverse propolis and its chemical composition on different *Candida* strains.

Propolis Origin	Fungal Species	Identified Compounds	Activity	Ref.
Brazil	*C. albicans* and *C. parapsilosis*	Caffeic acid, *p*-coumaric acid, cinnamic acid, aromadendrin, artepillin C	Showed fungicide (0.5%) action on different strains; *C. albicans* being the most sensitive, and *C. parapsilosis* being the most resistant	[92]
Brazil (propolis-based gels)	Vaginal candidiasis mouse model		Presented antifungal effect (1%) similar to clotrimazole cream	
Poland (50 different samples)	69 *C. albicans*, 10 *C. glabrata*, and 10 *C. krusei* clinical isolates	Caffeic acid, *p*-coumaric acid, ferulic acid, quercetin, pinobanksin, luteolin, kaempferol, apigenin, pinocembrin, acacetin, galangin, kaempferide, and naringenin	All samples showed high activity in the inhibition of biofilm formation by *C. glabrata* and *C. krusei* (0.04–1.25% *v*/*v*); inhibited yeast-to-mycelia morphological change and mycelial growth of *C. albicans* (0.16–0.31% *v*/*v*); propolis combined with fluconazole and voriconazole on *C. albicans* (0.5–512 μg/mL) was shown to have a clear synergism	[66]
Poland (from agricultural areas and Southern Poland)	*C. albicans* and *C. krusei*	Pinocembrin, chrysin, pinobanksin, apigenin, kaempferol, *p*-coumaric acid, ferulic acid, and caffeic acid	Samples from Southern Poland showed greater antifungal activity (2–16 mg/mL)	[43]

**Table 10 biology-10-00428-t010:** Effect of propolis on different species of *Trichophyton*.

Propolis Origin	Fungal Species	Activity	Ref.
Brazil (red and green propolis)	*T. rubrum*, *T. tonsurans*, and *T. mentagrophytes*	Both propolis were active on the strains, but the red propolis was more efficient than the green (256–1024 µg/mL)	[100]
Brazil (green propolis)	Clinical isolates of *T. rubrum* and *T. interdigitale* from onychomycosis cases	Propolis had the ability to decrease the cells in the preformed *Trichophyton* biofilm (0.044–0.088 mg/mL)	[101]
Brazil (topical green propolis treatment)	Sixteen patients with onychomycosis	Treatment showed a complete mycological and clinical cure of onychomycosis (10%)	

**Table 11 biology-10-00428-t011:** Antimalarial effect of propolis and its chemical composition.

Propolis Origin	Parasitic Species	Identified Compounds	Activity	Ref.
Cuba (20 different samples)	*P. falciparum*.	Liquiritigenin and lupeol	Three samples of propolis (0.2 μg/mL) shows activity against *P. falciparum*	[120]
Libya (12 different samples)		N.I.	All samples of propolis (1.65–53.6 μg/mL) showed antiprotozoal activity	[121]
Iran (four different samples)		Palmitic acid, stearic acid, pinocembrin, tectochrysin, and 4′,5-dihidroxy-7-methoxyflavanone	All samples of propolis presented antimalarial in vitro and in vivo activity at different concentrations (16.2–80 μg/mL)	[122]
Saudi Arabia	*P. chabaudi*-infected mice	N.I.	Considerably suppressed the parasitemia and demosntraed important effect in decresing anemic in infected mice (25–100 mg/Kg), reduced oxidative damage by enhancing the catalase function and the glutathione concentrations, and enhanced the quantity of TNF-α, IFN-γ, G-CSF, and GM-CSF.	[123]

N.I. = none identified.

**Table 12 biology-10-00428-t012:** Anti-trypanosome activity of different propolis and its chemical composition.

Propolis Origin	Parasitic Species	Identified Compounds	Activity	Ref.
Brazil	*T. cruzi*	N.I.	Activity on *T. cruzi* trypomastigotes.	[30]
		N.I.	Effective on the three forms of the parasite strongly inhibits infection levels promoting lysis of bloodstream trypomastigotes, diminishing the number of parasites in peritoneal macrophages and infected heart muscle cells (0.1–0.75 mg/mL)	[128]
		Caffeic acid, cinnamic acid, pantenoic acid, ferulic acid, linoleic acid, amyrin, pinostobin	In vitro effect against *T. cruzi* (0.4–1.4 mg/mL)	[129,130]
		3-prenyl-4-hydroxycinnamic acid and 2,2-dimethyl-6-carboxyethenyl-8-prenyl-2H-1-benzopyran.	Propolis (2.64 mg/mL) and its compounds (0.73–1.2 mg/mL) identified showed anti-trypanosome effects	[131]
Brazil and Bulgaria		N.I.	Both propolis (0.015–1.5 mg/mL) showed activity against *T. cruzi*, diminishing the infection and the intracellular replication of amastigotes; in epimastigotes, the main targets are the mitochondrion and reservosomes	[132]
Bulgaria (two different samples)		Caffeic acid, stearic acid, oleic acid, ferulic acid, coumaric acid, pinocembrin, chrysin, pinostrobin	Both samples of propolis had great inhibition effect mainly on *T. cruzi* epimastigotes (48.6–84.8 mg/mL); effect on trypomastigotes (160.5–1065.8 mg/mL) was similar to that of the reference drug	[133]
Bulgaria	*T. cruzi*-infected mice.	N.I.	Reduced parasitemia and showed no toxic hepatic or renal effect, decreased spleen mass, modulated the initial inflammatory reaction, favored a greater number of CD8^+^, and partially inhibited the increase in CD4 (50mg/Kg)	[134]
Brazil		N.I.	Reduced parasitemia enhanced the survival of the animals, and did not induce any hepatic, muscular lesion, or renal toxicity (25–300 mg/Kg)	[135]

N.I. = none identified.

**Table 13 biology-10-00428-t013:** Antileishmanial activity of various propolis and its chemical composition.

Propolis Origin	Parasitic Species	Identified Compounds	Activity	Ref.
Brazil (brown, green, and red propolis) and Portugal	*L. braziliensis*, *L. infantum*, and *L. amazonensis*	N.I.	All inhibited the growth of promastigotes of distinct parasitic strains and effectively reduced number of internalized amastigotes in infected murine macrophages (36–250 mg/mL)	[84,139,140,141]
Turkey (Three regions: Adana, Hatay, and Bursa)	*L. tropica* and *L. infantum*	Adana: cembrene; Hatay: chrysin; Bursa: cinnamyl cinnamate	All propolis (50–1000 μg/mL) showed good antileishmanial activity, but that of Bursa was the most effective	[142,143]
Cuba (20 different samples)	*L. infantum*	Amyrin, lupeol, and cycloartenol	All samples of propolis (3.2–22.2 μg/mL) presented antiprotozoal properties	[120]
Ecuador (three different samples)	*L. amazonensis*	Naringenin, sakuranetin, eupatolitin, and rhamnazin	All inhibited *L. amazonensis* growth, but the sample rich in flavonoids was the most effective (12.5–200 mg/mL)	[147]
Brazil (two different samples)	*L. amazonensis*, *L. braziliensis*, *L. chagasi*, and *L. major*	Minas Gerais: diethyl 2-methylsuccinate, cinnamic acid, pentanedioc acid, and hydrocinnamic acid; Ribeirao Petro: artepillin C, 4,5-dicaffeoylquinic acid, *p*-coumaric acid, and drupanin	Both showed great antileishmanial activity. Ribeirao Petro propolis exhibited a dose-dependent effect against promastigotes of *L. amazonensis* and controlled the parasite burden inside infected macrophages (2.8–229.3 μg/mL)	[148,149]
Brazil	*L. braziliensis*-infected mice	N.I.	Reduced growth by promoting morphologic alterations in promastigotes; in supernatants from liver cells and peritoneal exudate of mice pretreated with propolis and infected, increased TNF-α production was seen (5–100 μg/mL)	[150]
Brazil (green propolis)		N.I.	Decreases lesion development caused by *L. braziliensis* more than 75%, while the glucantime treatment showed a 57.7% decrease (10–250 μg/mL)	[151]

**Table 14 biology-10-00428-t014:** Activity of Brazilian propolis on different *Leishmania* infection models.

Propolis Origin	Substances	Infection Model	Activity	Ref.
Brazil	Nitric oxide	*L. braziliensis*-infected mice	The combination of propolis (5 mg/kg) with NO favored the healing, collagen synthesis, the function of macrophages and fibroblasts, reduced expression of proinflammatory and tissue damage markers	[152]
Brazil (green propolis)	Liposomal meglumine antimoniate	*L. infantum*-infected mice	Reduced the parasitic burden in the liver, without damaging kidney, liver, spleen, and heart (500 mg/kg)	[153]

**Table 15 biology-10-00428-t015:** Effect of different propolis on *G. lamblia* in in vitro and in vivo models.

Propolis Origin	Parasitic Species	Activity	Ref.
Mexico (Sonoran Desert, region Ures)	*G. lamblia*	Inhibitory activity against *G. lamblia* in a dose-dependent manner (63.8 μg/mL)	[159]
Brazil		Effectively eliminated trophozoites of *G. lamblia* (125 μg/mL)	[160,161]
Egypt	*G. lamblia*-infected immunodeficient mice	Reduces infection, enhanced IFN-γ serum level and CD4+:CD8+ T cell ratio. Co-administration of propolis and metronidazole had remarkable activity in controlling the parasite number. Favors intestinal homeostasis and the histological integrity (NS)	[162]

NS: not specified.

**Table 16 biology-10-00428-t016:** Anthelmintic activity of propolis and its chemical composition.

Propolis Origin	Parasitic Species	Identified Compounds	Activity	Ref.
Egypt	*F. gigantica*	Diprenyl-dihydrocoumaric acids, coumarate esters, ferulate esters, hydroxy acetophenones, furanon derivative, furofuran lignans, benzofuran lignans, and valeric acids derivatives	Alteration in the architecture, inhibitory activity on the viability and hatchability of immature helminths (10–800 μg/mL)	[170,171]
	*S. mansoni*-infected mice	N.I.	Propolis administration (500 mg/kg) not eliminated the worms of infected mice, but significantly reduced the hepatic granuloma number, hepatic, splenic and plasma MPO activity, as well the liver and thymus NO levels; also, regulation of plasma antioxidant proteins evidenced by decrease in MDA and normalization of GSH	[172]
Turkey	*E. granulosus*	N.I.	Propolis (1 µg/mL) killed all the protoscoleces	[173]
Brazil (essential oil of red propolis)	*T. cati*	N.I.	Have 100% larvicidal effect after treatment (1 µg/mL) of 48 h and can suppress the ability of the treated *T. cati* larvae to infect the mice	[174]
Libya (five different samples)	*T. spiralis*	Cycloartanol, mangiferolic acid, agathadiol, isocupressic acid, isoagatholal	All propolis samples (4.7–59.3 μg/mL) showed moderate activity	[175]

N.I., none identified; MPO, myeloperoxidase; MDA, malondialdehyde; GSH, glutathione.

**Table 17 biology-10-00428-t017:** Antiviral activity of different samples of propolis.

Propolis Origin	Viral Species	Activity	Ref.
Czech Republic (aqueous and ethanolic extract)	HSV-2	Both propolis (0.0005–0.005%) decreased the infection and exhibited a concentration- and time-dependent antiviral effect; high antiviral effect when viruses were pretreated prior to infection	[188]
United States, Brazil, and China	HIV-1.	Three propolis inhibited viral expression in CD4+ lymphocytes and microglial cell in a concentration-dependent manner; propolis from United States suppressed cell fusion in cultures of CD4+ lymphocytes (0.8–66.6 μg/mL)	[189]
Brazil (13 different samples)	Influenza virus.	Four samples had anti-influenza virus effect in vitro (10–149.2 μg/mL)	[190]
	Influenza virus-infected mice.	Only one propolis sample (10 mg/mL) effectively prolonged the lifetimes of infected mice; anti-influenza effectiveness of propolis in mice was dose-dependent	
Brazil (13 different samples)	HSV-1-infected mice	The three propolis had direct anti-HSV-1 effects, stimulated immunological effect on intradermal HSV-1 infection in mice (0.4, 2, and 10 mg/mL)	[191]

**Table 18 biology-10-00428-t018:** Anti-HSV-1 and HSV-2 activity of several propolis and its chemical composition.

Propolis Origin	Parasitic Species	Identified Compounds	Activity	Ref.
Canada	HSV-1 and HSV-2	Benzoic acid, cinnamic acid, vanillic acid, *p*-coumaric acid, ferulic acid, caffeic acid, palmitic acid, oleic acid, pinocembrin, pinobanksin, chrysin, galangin, isosakuranetin, alpinone, kaempferol, pinostrobin chalcone, and pinocembrin chalcone	Presents a pronounced viricidal effect and interfered with virus adsorption (0.1 mg/mL)	[194]
Turkey (south)		Gallic acid, (±)-catechin, caffeic acid, syringic acid, epigallocatechin, *p*-coumaric acid, trans-ferulic acid, trans-isoferulic acid, myricetin, trans-cinnamic acid, benzoic acid, daidzein, luteolin, pinobanksin, (±)-naringenin, apigenin, kaempferol, chrysin, pinocembrin, galangin, and CAPE	Suppressed the replication of HSV-1 and HSV-2; inhibited HSV-1 replication following 24 h of incubation and effect on HSV-2 at 48 h following incubation; decreased the number of viral copies; activity similar to that of acyclovir; a synergism of the propolis and acyclovir combined on HSV-1 and HSV-2 replication compared with acyclovir alone (25–3200 μg/mL)	[185]
Czech Republic (aqueous and ethanolic propolis)	HSV-1	Caffeic acid, *p*-coumaric acid, benzoic acid, galangin, pinocembrin, and chrysin	Both samples presented high anti-HSV-1 effect in cells pretreated prior to viral infection; galangin and chrysin were the most bioactive compounds; however, the propolis had higher antiherpetic effects than single isolated constituents (1%)	[187]

**Table 19 biology-10-00428-t019:** Antiherpetic activity of Central European propolis in patient studies.

Propolis Origin	Model	Activity	Ref.
Central Europe (lip cream with propolis)	Patients with herpes labialis	Lip cream with propolis (0.5%) produced a better effect than acyclovir in the treatment of patients with herpes labialis in the vesicular phase	[200]
Central Europe (constituent of a lotion)	Herpes zoster in patients	Improvement in pain and healing of lesions were better and quicker with treatment of the propolis lotion (0.5%); approximately 50% of propolis-treated patients had no injuries on day 14 and the formation of new vesicles was suppressed	[201]

## Data Availability

Not applicable.

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
