# Peer review of "Effects of Propolis on Infectious Diseases of Medical Relevance"

_biology, 2021, doi:10.3390/biology10050428_

Round 1

Reviewer 1 Report

The present review is well structured and contains a lot of information about propolis. I have some very minor suggestions for the authors to imporve their work:

-after the first time, the name of the genus should appear in contract form (e.g.  Candida genus is repeated several times.. but it could be indicated as C.)

-one or more images resuming the potential activities of the propolis should be reported in the MS to allow readers to better focalise the huge amount of data present in literature

-I would suggest to the authors to open, in the conclusions, new perspectives about the potential use of propolis. In particular, the most recent issue of the existence of a cross-kingdom bioactivity performed by plant microRNA introduced by diet or pharmaceutical products and able to modulate gene expression of the consumers.. for beehive products (which could suggest the possibility that even propolis can share the same properties), I would suggest the following articles: PLoS One12(2), e0172981;  PLoS genetics, 2017, 13.8: e1006946.

Author Response

Reviewer 1's corrections:

Point 1. after the first time, the name of the genus should appear in contract form (e.g.  Candida genus is repeated several times.. but it could be indicated as C.)

 Regarding the comment:

Corrections to the candida genus were made throughout the section.

Point 2. One or more images resuming the potential activities of the propolis should be reported in the MS to allow readers to better focalise the huge amount of data present in literature

 Regarding the comment:

In line 20, a graphic abstract was added for the reader to locate the context of the topics to be addressed in the review.

Point 3. I would suggest to the authors to open, in the conclusions, new perspectives about the potential use of propolis. In particular, the most recent issue of the existence of a cross-kingdom bioactivity performed by plant microRNA introduced by diet or pharmaceutical products and able to modulate gene expression of the consumers.. for beehive products (which could suggest the possibility that even propolis can share the same properties), I would suggest the following articles: PLoS One12(2), e0172981;  PLoS genetics, 2017, 13.8: e1006946.

 Regarding the comment:

The suggested studies are of great relevance in current research on bee products. They were added in the conclusions section as a perspective for future research.

We add:

In line 1071: “Finally, a new perspective to consider in future research is to investigate the presence and function of microRNAs (miRNAs) in propolis. Recent studies have proposed that the miRNAs present in honey from plants visited by bees during their collection, that these molecules could play a determining role in the development of larvae. The finding of these molecules could be surprising related to beneficial effects on the health of consumers of this bee product. The identification of miRNAs in propolis would be crucial to understand and explain many of its biological and medicinal activities and that these activities are currently attributed mainly to compounds such as flavonoids and terpenes. miRNAs in bee products can be subject of various investigations and their clinical applications could generate new treatments based on nutritional supplements with various specific benefits for health [205,206].”

Reviewer 2 Report

The manuscript is interesting and highlights the pharmacological properties of propolis, a drug used for a very long time

The authors have not described the role of propolis in Covid-19, I suggest looking at these manuscripts and others to complete a paragraph.

Ripari N, Sartori AA, da Silva Honorio M, Conte FL, Tasca KI, Santiago KB,

Sforcin JM. Propolis antiviral and immunomodulatory activity: a review and

perspectives for COVID-19 treatment. J Pharm Pharmacol. 2021 Mar

6;73(3):281-299. doi: 10.1093/jpp/rgaa067. PMID: 33793885; PMCID: PMC7928728.

Ali AM, Kunugi H. Propolis, Bee Honey, and Their Components Protect against

Coronavirus Disease 2019 (COVID-19): A Review of In Silico, In Vitro, and

Clinical Studies. Molecules. 2021 Feb 25;26(5):1232. doi:

10.3390/molecules26051232. PMID: 33669054; PMCID: PMC7956496.

Berretta AA, Silveira MAD, Cóndor Capcha JM, De Jong D. Propolis and its

potential against SARS-CoV-2 infection mechanisms and COVID-19 disease: Running

title: Propolis against SARS-CoV-2 infection and COVID-19. Biomed Pharmacother.

2020 Nov;131:110622.

I also suggest to the authors a graphical abstract, because the result is more direct

Furthermore I also suggest making a small paragraph on the chemical constituents of propolis and also the role they play (eg. galangin)

In the Discussion, the Authors should highlight the possible clinical significance of their findings

Author Response

Reviewer 2's corrections

Point 1. The authors have not described the role of propolis in Covid-19, I suggest looking at these manuscripts and others to complete a paragraph.

Ripari N, Sartori AA, da Silva Honorio M, Conte FL, Tasca KI, Santiago KB, Sforcin JM. Propolis antiviral and immunomodulatory activity: a review and perspectives for COVID-19 treatment. J Pharm Pharmacol. 2021 Mar 6;73(3):281-299. doi: 10.1093/jpp/rgaa067. PMID: 33793885; PMCID: PMC7928728.

Ali AM, Kunugi H. Propolis, Bee Honey, and Their Components Protect against Coronavirus Disease 2019 (COVID-19): A Review of In Silico, In Vitro, and Clinical Studies. Molecules. 2021 Feb 25;26(5):1232. doi:10.3390/molecules26051232. PMID: 33669054; PMCID: PMC7956496.

Berretta AA, Silveira MAD, Cóndor Capcha JM, De Jong D. Propolis and its potential against SARS-CoV-2 infection mechanisms and COVID-19 disease: Running title: Propolis against SARS-CoV-2 infection and COVID-19. Biomed Pharmacother. 2020 Nov;131:110622.

Regarding the comment: 

The suggested studies on COVID-19 broaden the focus of this review and enriched the discussion in this section.

We add:

Line 1030: “The current situation related to COVID, which compromised all health systems, makes it necessary to search for therapies that prevent or mitigate the complications of this disease. natural products such as propolis are an interesting option in the search of complete therapies. some recent research mentions the potential benefits of using propolis against this disease. These studies are based on previously reported activities against other viruses and on in silico models that allow predictions of activities against this virus. These research focus mainly on reported bioactive compounds in the different propolis, they include antiviral activities that could be applied against SARS-CoV-2 or immunomodulatory effects that would reduce the symptoms of the disease. Some of the clearest examples are quercetin, one of the most abundant and consumed flavonoids in the diet. Quercetin has shown to inhibit the replication cycle of the virus, since it reduces the functioning of the main protease (Mpro) and S protein of SARS-CoV-2. CAPE, one of the main components of many propolis, is able to inhibit the Transmembrane protease serine 2 (TMPRSS2), Angiotensin-converting enzyme-related carboxypeptidase (ACE-II), and Mpro, these molecules are crucial for access and replication viral of SARS-CoV-2 in cells. another interesting compound is the rutin that reducedthe function of S protein, ACE-II, and others non-structural proteins of SARS-CoV-2. These flavonoids are also able to regulate JAK / STAT-mediated signaling, the production of ROS, NO, pro and anti-inflammatory cytokines, avoid cytokine storm. They even reduce the risk of comorbidities that complicate the betterment of patients with COVID-19 [202-204]. Propolis and its bioactive compounds open new ways for future works that describe in detail the effects of these on SARS-Cov-2 and be able to apply them as a complementary therapy in clinical studies.”

Point 2. I also suggest to the authors a graphical abstract, because the result is more direct

Regarding the comment: 

A graphic abstract was added for the reader to locate the context of the topics to be addressed in the review

Point 3. Furthermore I also suggest making a small paragraph on the chemical constituents of propolis and also the role they play (eg. galangin)

Reply:

The aim of the review is to exalt the benefits of propolis on human health. The suggestion is very interesting, however, we consider that the great variety of the chemical composition of propolis makes it difficult to decide which are the most important compounds of this natural product. furthermore, delving into the discussion of bioactive compounds could mislead the reader as to the purpose of the review. If you think it is necessary, we are ready to write it in the second reply.

Point 4. In the Discussion, the Authors should highlight the possible clinical significance of their findings

Regarding the comment: 

We add:

Line 381: “As we already mentioned, resistance to antibiotics is a serious health problem since it makes it difficult to properly treat several diseases of bacterial origin. The documented effects of propolis and its derivatives on bacteria such as MRSA make them ideal candidates for clinical studies in order to evaluate their effectiveness on antbiotic-resistant bacterial diseases. The clinical application of propolis shouldn’t focus on the substitution of antibiotics but on complementing and improving the efficacy of these when co-administered.”

Line 613: “so it is important to realize more clinical trails to support effectiveness of propolis, in addition to implement trials focused on evaluating toxicity to determine a standardized dose of propolis that is safe for consumption and application in humans, as well as study the components of each propolis used. These studies could give scientific support to natural products widely used as a therapeutic alternative in rural communities and in developing nations.”

Line 843: “It should be noted that the implementation of clinical studies is necessary to support the antiparasitic activity of propolis, as well as conducting research focused on the combined treatment of propolis with different drugs used to treat parasitic infections in humans and this aspect find more effective complementary in order to be able to reduce the dose and toxicity of the drugs currently implemented.”

Line 1049: “These flavonoids are also able to regulate JAK / STAT-mediated signaling, the production of ROS, NO, pro and anti-inflammatory cytokines, avoid cytokine storm. They even reduce the risk of comorbidities that complicate the betterment of patients with COVID-19 [202-204]. Propolis and its bioactive compounds open new ways for future works that describe in detail the effects of these on SARS-Cov-2 and be able to apply them as a complementary therapy in clinical studies . ”

Line 1071: “Another relevant aspect are clinical tests with propolis, bee products or other natural products are scarce but necessary. Since from products used in traditional medicine, modern medicine has obtained compounds such as taxol, valproic acid, polycarpine, ephedrine, digoxin, acetylsalicylic acid, just to name a few. The therapeutic uses and applications of natural products and their derivatives are promising in the search for new treatments, so clinical studies against diseases caused by microbes resistant to drugs or treated with toxic agents should be a priority in future clinical research.”

Reviewer 3 Report

The manuscript by Rivera-Yañezand co-workers (biology-1212315) is a review dealing with the biological activities of propolis.

The topic is of great interest, the manuscript is well written and, in my opinion, suitable for publication in Biology.

However, the main problem with this review is that numerous research papers are mentioned where the bioactivity of propolis has been evaluated, without reporting quantitative data either in the text or in the tables.

I recommend improving the manuscript by reporting the quantitative data regarding the concentrations of the main bioactive components in propolis of different origin and the biological activities compared to those of the drugs currently in use against the pathogens of interest. This comparison is essential to evaluate the potential of propolis in the prevention and treatment of diseases, as well as its use to improve the effectiveness of traditional therapies.

In the attached pdf file some small changes to be made to the manuscript are indicated.

Author Response

Reviewer 3's corrections

The topic is of great interest, the manuscript is well written and, in my opinion, suitable for publication in Biology.

However, the main problem with this review is that numerous research papers are mentioned where the bioactivity of propolis has been evaluated, without reporting quantitative data either in the text or in the tables.

I recommend improving the manuscript by reporting the quantitative data regarding the concentrations of the main bioactive components in propolis of different origin and the biological activities compared to those of the drugs currently in use against the pathogens of interest. This comparison is essential to evaluate the potential of propolis in the prevention and treatment of diseases, as well as its use to improve the effectiveness of traditional therapies.

In the attached pdf file some small changes to be made to the manuscript are indicated.

Reply:

The corrections and suggestions were made.

Line 90: Staphylococcus

Line 149: p-coumaric

Line 153: p-coumaric

Line 226 and 227: Streptococcus

Line 257: Th

Line 281: p-coumaric

Line 290: p-coumaric

Line 350: p-coumaric

Line 365: p-coumaric

Line 411 and 412: Candida

Line 224: Candida

Line 235 and 236: Candida

Line 529-532: Trichophyton

Line 588: Aspergillus

The requested concentrations were added.

Line 130 (Table 1)

Line 170 (Table 2)

Line 217 (Table 3)

Line 252 (Table 4)

Line 304 (Table 5)

Line 343 (Table 6)

Line 373 (Table 7)

Line 468 (Table 8)

Line 512 (Table 9)

Line 555 (Table 10)

Line 660 (Table 11)

Line 702 (Table 12)

Line 774 (Table 13)

Line 794 (Table 14)

Line 883 (Table 15)

Line 894 (Table 16)

Line 961 (Table 17)

Line 978 (Table 18)

Line 1027 (Table 19)

Round 2

Reviewer 3 Report

In my opinion the Authors have responded satisfactorily to the Referees and the manuscript is now ready for publication